# The α-globin super-enhancer acts in an orientation-dependent manner

Mira T. Kassouf [1] ✉, Helena S. Francis [2,6], Matthew Gosden [2,6], Maria C. Suciu[2], Damien J. Downes [2], Caroline Harrold [2], Martin Larke[2], Marieke Oudelaar [3], Lucy Cornell [1], Joseph Blayney[1], Jelena Telenius[2], Barbara Xella[2], Yuki Shen [2], Nikolaos Sousos [2], Jacqueline A. Sharpe[2], Jacqueline Sloane-Stanley[2], Andrew J. H. Smith[4], Christian Babbs [1], Jim R. Hughes [2] & Douglas R. Higgs [1,5] ✉

Individual enhancers are defined as short genomic regulatory elements, bound by transcription factors, and able to activate cell-specific gene expression at a distance, in an orientation-independent manner. Within mammalian genomes, enhancer-like elements may be found individually or within clusters referred to as locus control regions or super-enhancers (SEs). While these behave similarly to individual enhancers with respect to cell specificity, distribution and distance, their orientation-dependence has not been formally tested. Here, using the α-globin locus as a model, we show that while an individual enhancer works in an orientation-independent manner, the direction of activity of a SE changes with its orientation. When the SE is inverted within its normal chromosomal context, expression of its normal targets, the α-globin genes, is severely reduced and the normally silent genes lying upstream of the α-globin locus are upregulated. These findings add to our understanding of enhancer-promoter specificity that precisely activate transcription.

Since the first description of enhancers over 40 years ago, these elements have been defined as segments of DNA containing a high density of transcription factor binding sites that boost transcription of their target genes over large distances. Depending on the transcription factors that they bind, they regulate tissue-specific expression[1]. Furthermore, enhancers determine when genes are switched on during development, lineage-specification and differentiation[2]. They are said to do so whether located upstream or downstream of the target transcriptional unit and in an orientation-independent manner[3–5]. This definition implies that enhancers and promoters act as freely diffusing genomic elements. While this may be true in simple experimental assays, the situation within a natural chromosomal environment is more complex.

In mouse and human, there are more than 500,000 elements marked with chromatin signatures indicative of enhancer activity[1,6,7]. However, only a proportion of these elements act as enhancers in stringent assays[8,9]. Recently, we have shown that some enhancer-like elements have no activity on their own but they significantly increase the activity of classical enhancers: such enhancer-like elements have been referred to as "facilitators"[10]. Genome-wide studies have shown that enhancer-like elements (enhancers and facilitators) may occur in clusters, referred to as locus control regions[11] or super-enhancers (SEs)[12]. Whether acting as single elements or clusters it is thought that enhancers form a non-membrane bound nuclear compartment containing a high concentration of transcription factors and co-factors, sometimes

[1]Gene Regulation Laboratory, MRC Weatherall Institute of Molecular Medicine, John Radcliffe Hospital, OX3 9DS Oxford, UK. [2]MRC Molecular Haematology Unit, MRC Weatherall Institute of Molecular Medicine, John Radcliffe Hospital, OX3 9DS Oxford, UK. [3]Max Planck Institute for Multidisciplinary Sciences, 37077 Gottingen, Germany. [4]Institute for Regeneration and Repair, MRC Centre for Regenerative Medicine, University of Edinburgh, Edinburgh, Scotland EH16 4UU, UK. [5]Chinese Academy of Medical Sciences Oxford Institute, University of Oxford, OX3 7BN Oxford, UK. [6]These authors contributed equally: Helena S. Francis, Matthew Gosden. ✉e-mail: mira.kassouf@imm.ox.ac.uk; doug.higgs@imm.ox.ac.uk

referred to as a transcriptional hub[13,14]. When the transcriptional hub and its cognate promoters come into proximity, this is thought to enable the recruitment of the pre-initiation complex to the promoter, and to expedite various stages of the transcriptional cycle[15]. In the simplest model, it might be expected that, as in the classical definition of enhancers, this arrangement should work equally well whether the enhancers lie upstream or downstream of their target gene(s) and there are many examples of both configurations in the genome[16,17].

The importance of the orientation of enhancers with respect to the genes they control within their normal chromosomal context has not been widely tested or examined in detail. Here we have addressed this issue using the well characterised mouse α-globin cluster as a model. The entire α-globin cluster lies within a ~165 kb TAD (topologically associating domain) present in all tested cell types. Specifically, in erythroid cells, a ~65 kb erythroid-specific sub-TAD develops as cells progress through terminal differentiation and the α-globin genes are transcribed at increasingly higher levels[18]. The sub-TAD is thought to form as a result of cohesin-mediated loop-extrusion delimited by largely convergent CTCF insulator elements. Within the 65 kb sub-TAD, a set of five erythroid-specific enhancer-like elements (R1, R2, R3, Rm and R4) regulate an embryonic ζ-globin gene (Hba-x) and a pair of almost identical adult α-globin genes (Hba-1 and Hba- 2). These elements are located ~8–31 kb 5′ (upstream) of Hba-x, with four of them (R1, R2, R3 and Rm) embedded in introns of the adjacent widely expressed gene Nprl3 (Fig. 2A). Also present within the sub-TAD are two θ-globin genes (Hbθ-1 and Hbθ-2) of unknown function. This cluster of enhancer-like elements fulfil the definition of a SE[19] and we have recently shown that whereas R1 and R2 are classical enhancers, R3, Rm and R4 act as facilitators[10].

Here we have addressed the importance of the orientation of the SE within the sub-TAD in activating gene expression. We have found that the orientation of the major enhancer (R2) within the otherwise intact SE has no effect on α-globin transcription. By contrast, reversal of the entire SE redirects activity from the α-genes to the genes which normally lie upstream of the α-globin cluster, showing that, in contrast to single enhancers, the multi-partite α-globin SE acts in an orientation-dependent manner.

## Results

### Inversion of one major enhancer element has no effect on α-globin expression

To address the importance of the orientation of individual enhancer-like elements we used the mouse α-globin cluster as a model (summarized in Supplementary Fig. 1). The α-globin cluster lies within a 165 kb TAD surrounded by widely expressed genes. When active, in terminally differentiating erythroid cells, all five elements of the α-globin SE come into close proximity to the duplicated α-globin gene promoters in a ~65 kb sub-TAD flanked by closely apposed, largely convergent CTCF elements[20] (Supplementary Fig. 1a). Expression and chromatin state of the α-globin locus (Supplementary Fig. 1b), and the widely-expressed flanking genes, can be accurately assessed in mouse models with easily accessible erythroid cells, and in well-validated mouse embryonic stem cell (mESC) lines differentiated along the erythroid lineage[19,21,22].

Although it is widely accepted that mammalian enhancers act in an orientation-independent manner[3], this has been formally tested at relatively few complex endogenous loci in vivo[1]. To test this within the context of a well-defined regulatory domain, we examined the effect of inverting the strongest constituent enhancer (R2) on expression of its target genes. We modified mESC lines and studied the effect of inverting R2 on expression of the α-like globin genes in erythroid differentiation[22] (Fig. 1a, EB-derived Ery). Embryoid body-derived erythroid cells (EB-derived Ery) behave as primitive (embryonic) cells when comparing their chromatin accessibility and gene expression profiles to primary primitive erythroblasts obtained from E10.5 peripheral blood and definitive erythroid cells (Ter119+ spleen red blood cells)[22] (Fig. 1a). Most importantly, EB-derived erythroid cells express

both embryonic ζ– (Hba-x) and adult α– (Hba-a1/2) globin, both under varying degrees of control from the α-globin SE, as described in primary primitive and definitive erythroid cells[19,23].

We initially created a mESC model in which the R1 enhancer element was deleted from both alleles (Fig. 1b ii, ΔR1). As reported previously[19], the pattern of chromatin accessibility at the remaining enhancers is unchanged in the presence or absence of R1 in primary erythroid cells. As in engineered mice, removal of R1 in mESCs led to a ~40–50% downregulation of α-globin mRNA in EB-derived erythroid cells (Fig. 2a, b, ΔR1). Therefore, we generated ΔR1 EB-derived erythroid cells in which α-globin expression almost entirely depends on R2 and its facilitators (R3, Rm and R4), providing a model in which to study the importance of the orientation of the single major classical enhancer (R2). To examine the effect of R2 orientation in this setting, we compared erythropoiesis, chromatin accessibility and α-globin expression in EB-derived erythroid cells from mESCs in which R1 was deleted and R2 was either in its native orientation (ΔR1) or inverted (ΔR1-R2[INV]) on both alleles (Fig. 1b ii, iii): the facilitators remained unchanged. Erythropoiesis, chromatin accessibility and α-globin expression (Hba-a1/2) appeared indistinguishable in two independently derived clones of each of these mESC lines, showing that a single major enhancer (R2) acts in an orientation-independent manner in the context of the otherwise intact α-globin regulatory domain (Fig. 2a, b, ΔR1-R2[INV]). Furthermore, expression of Hba-x, Nprl3 and erythroid control genes (CD71 and pb4.2) was also unchanged in ΔR1-R2[INV] erythroid cells (Fig. 2b). We therefore concluded that the orientation of a single classical enhancer (R2) has no effect on its target genes' expression.

### Inversion of the SE down-regulates α-globin expression in a mESC model

The order of the α-globin enhancer-like elements within the SE has been conserved throughout at least 70 million years of evolution[24,25]. We have shown above that the R2 element, at its native position in the genome, exerts equal effect in either orientation on α-globin expression, in agreement with the established enhancer biology paradigm. However, the question of whether a cluster of linked enhancer-like elements within a SE also acts in a similar orientation-independent manner is not clear. To address this, we asked whether inverting the entire SE with respect to the α-globin promoters affects gene expression (Fig. 1b i, v, grey shaded area). To do this we inverted a region of the mouse α-globin locus containing all five enhancers within the confines of the highly conserved syntenic regulatory domain (5′-Rhbdf1-Mpg-Nprl3-Hba-x- Hba-a1/2-Hba-q1/2-3′). Since disruption of the Mpg or Nprl3 genes would heighten susceptibility to genotoxicity and lead to developmental abnormalities, respectively[26,27], we inverted a 50 kb region containing Mpg, Nprl3, and the α-globin regulatory elements in their entirety. To achieve this, convergent LoxP sites were integrated at flanking insertion sites devoid of chromatin features associated with regulatory elements (Supplementary Fig. 1b, shaded orange bars). Inversion of this segment of DNA in mESCs occurred upon expression of Cre recombinase (Supplementary Fig. 1c, and Methods). Subsequently, all selectable markers were removed using site-specific recombinases (Supplementary Fig. 1c, and Methods). The resulting cell line was termed SE[INV]. Given the extensive genome engineering performed to generate this model, we have examined and shown using Bionano Optical Genome Mapping that the integrity of the DNA sequence within and in the flanking regions of the inversion as well as genome-wide is preserved (Supplementary Fig. 2a). It is important to note that the overall change in distance between the major enhancers (R1 and R2) and the α-globin promoters is negligible (~5 kb) in the inversion. The inversion has no impact on chromatin accessibility over the cluster of enhancers (Fig. 3a). CTCF binding across the locus appeared unaltered (Fig. 3a) although the interactions of the enhancers change such that 5′ interactions are favoured at the expense of 3′ interactions (Fig. 3b). Interestingly, expression of both

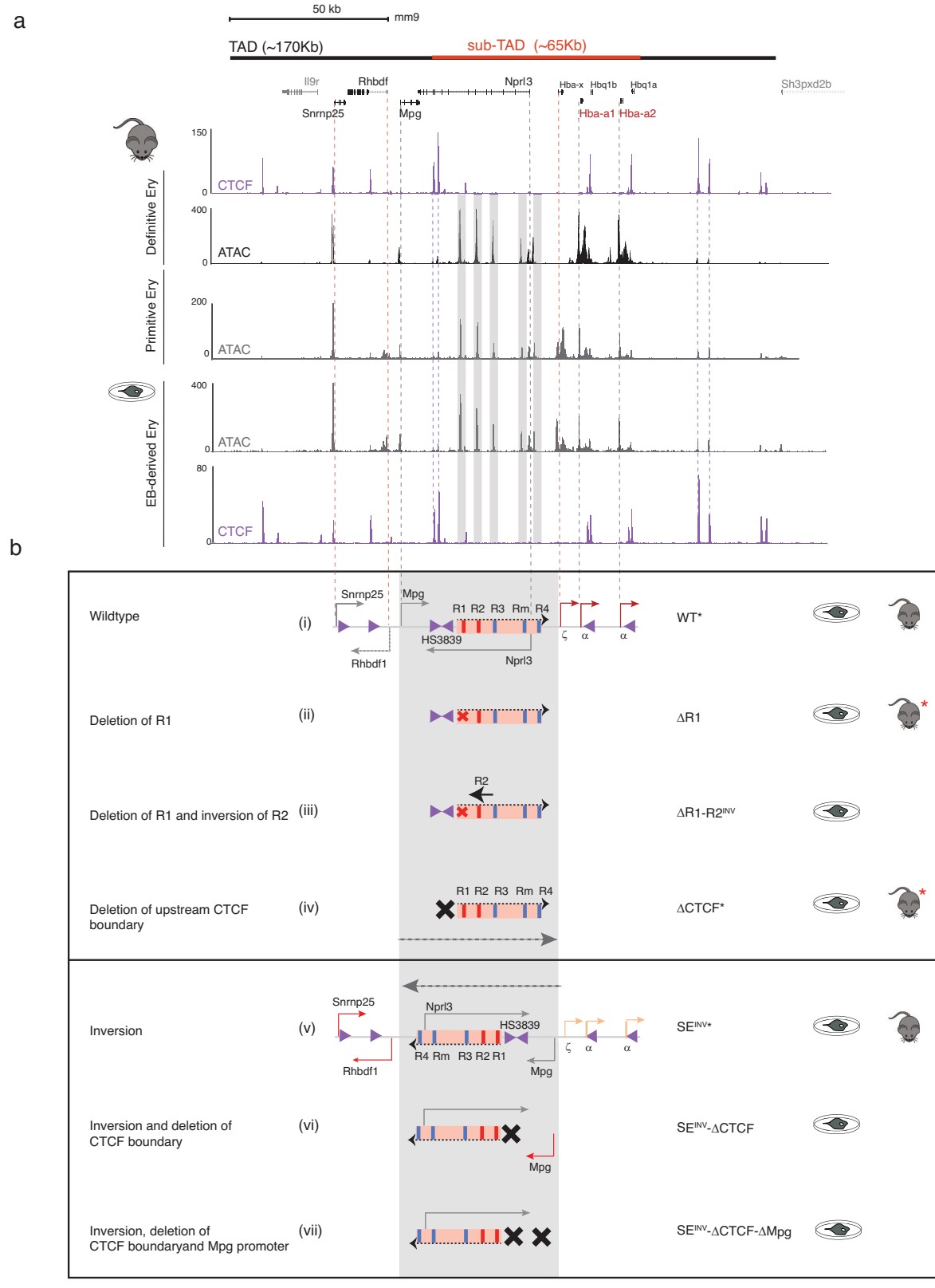

embryonic (*Hba-x*) and adult (*Hba-a1/2*) α-globin is reduced to ~20% of normal in EB-derived SE[INV] erythroid cells whilst expression of the surrounding genes (*Rhbdf1* and *Snrnp25* in their endogenous location and *Mpg* in the repositioned location) is increased (Fig. 3c). Despite preserving all the *cis*-elements necessary for full α-globin expression, inverting the SE significantly downregulates expression of its target genes and now upregulates genes lying upstream.

## Inversion of the SE leads to an α−thalassaemia phenotype in homozygous mice

To investigate the effect of the inversion on α-globin expression in primary cells, at all stages of development, mESCs heterozygous for the inversion were used to generate a mouse model harbouring the inversion described above (Fig. 1b v, Supplementary Fig. 1c). Mice heterozygous (WT/SE[INV]) or homozygous (SE[INV]/SE[INV]) for the inversion

**Fig. 1 | The erythroid systems and mouse models developed and analysed in this study. a** Top, solid black and red bars represent the α-globin TAD and sub-TAD respectively. RefSeq and coordinates (mm9) chr11:32,080,000–32,250,000 bp. Below, CTCF occupancy (ChIP-seq) and open chromatin (ATAC-seq) profiles in primary erythroid cells (with the 'mouse' graphic) derived from the definitive and primitive lineages as well as EB-derived erythroid cells (with 'a cell in a dish' graphic). **b** Top box contains schematics representing (i) the wildtype α-globin locus with arrows marking the embryonic (ζ) and adult α-globin genes (α) (in red) and flanking genes (*Nprl3, Mpg, Rhbdf1, Snrnp25,* in grey) and pointing in the direction of their expression. Orange box represents the SE: red bars mark the enhancers (R1, R2), blue bars the facilitators (R3, Rm, R4). Purple arrows indicate the CTCF-binding sites, the arrow direction represents the CTCF site orientation, and the α-globin tested boundary elements labelled HS3839. The grey shaded area highlights the region inverted in this study. Schematics represent (ii) R1 deletion mutant (red cross, ΔR1), (iii) R1 deletion (red cross) and R2 inversion (inverted black arrow) mutant (ΔR1-R2$^{INV}$), (iv) HS3839 boundary deletion mutant (black cross, ΔCTCF). Dashed grey arrow indicates the natural orientation of the SE pointing towards the α-globin genes. The lower box contains the inversion models (SE$^{INV}$) as indicated by the dashed grey arrow pointing away from α-globin genes and towards *Rhbdf1* and *Snrnp25* genes. (v) With the exception of the orientation of the grey-shaded area, and the colour of the arrows representing changes in levels of expression (orange indicating down regulation of the α-globin genes and red arrows indicating upregulation of *Rhbdf1* and *Snrnp25*), the annotation of the elements remains the same as in (i). (vi) The SE$^{INV}$ model harbouring the HS3839 CTCF boundary deletion (black cross) and red arrow indicating *Mpg* gene upregulation (SE$^{INV}$-ΔCTCF). (vii) The SE$^{INV}$-ΔCTCF model harbouring *Mpg* gene knock-out (second black cross, SE$^{INV}$-ΔCTCF-ΔMpg). Black stars (WT*, ΔCTCF*, SE$^{INV}$*) indicate the models in which *Mpg* gene was also deleted and analysed (ΔMpg, ΔCTCF-ΔMpg, SE$^{INV}$-ΔMpg). Graphics of 'cell in a dish' and 'mouse' represent mESC in vitro culture and in vivo mouse models respectively. Red stars indicate mouse models previously published.

survived to adulthood, bred normally, and were observed at the expected Mendelian ratios (Supplementary Fig. 2b). As in the mESC SE$^{INV}$ model, chromatin accessibility and modifications at the enhancer cluster were unchanged in primary erythroid cells compared to those derived from WT mice (Fig. 4a). However, ATAC and H3K4me3 chromatin peaks were reduced at the α-globin promoters (Fig. 4a). Newborn and adult mice harbouring the inversion were anaemic, showed a significant reduction in the α/β globin RNA ratio (Fig. 4b), presented with splenomegaly (Fig. 4c, Supplementary Fig. 2c) and increased levels of immature red cells (reticulocytes) in their peripheral blood (Fig. 4c). The appearance of the blood films and red cell indices (Fig. 4c) indicated a typical hypochromic microcytic anaemia associated with α-thalassaemia. Flow cytometry of the erythroid populations derived from the bone marrow of homozygous SE$^{INV}$ mice showed no block in erythroid differentiation but a modest expansion of immature populations due to stress erythropoiesis caused by the α-thalassaemia (Supplementary Fig. 2d).

As the in vitro mESC model produces EB-derived primitive-like red cells[22], we also harvested E10.5 primitive red cells derived from SE$^{INV}$ mice for comparison. The effect of the inversion in the primary primitive cell context recapitulated the reduced embryonic *Hba-x* and adult α-globin (*Hba1/2*) gene expression seen in the in vitro model (Supplementary Fig. 3b). E10.5 embryonic erythroid cells show a chromatin accessibility profile similar to that observed in primary SE$^{INV}$ definitive (Fig. 4a) and EB-derived SE$^{INV}$ erythroid cells (Fig. 3a) marked specifically by a reduced accessibility over the globin promoters and increased accessibility over the *Rhbdf1* and *Snrnp25* genes (Supplementary Fig. 3a). Irrespective of the stage of erythroid development (embryonic or adult), inverting the α-globin SE significantly downregulates both *Hba-x* and *Hba1/2* gene expression.

### The inversion changes the SE chromatin conformation and activity at the α-globin locus

We next asked how inversion of the SE affects chromatin interactions in the α-globin regulatory domain. Using the NG Capture C technique[28], the self-interacting domain, referred to as a sub-TAD, is normally observed as a ~65 kb region of increased chromatin interactions, which is formed specifically both in EB-derived and mouse-derived erythroid cells (Figs. 3b, 4e, WT tracks). This sub-TAD extends across the entire α-globin cluster with preferential interactions occurring between all of the α-globin enhancer-like elements and promoters[20,22,23,28]. The erythroid-specific, self-interacting domain is flanked by largely convergent CTCF/cohesin binding sites some of which can act as domain insulators and restrict the range of interactions of the enhancers with the flanking genes (Supplementary Fig. 4, CTCFH38 and CTCFH44 viewpoints, WT tracks). The inversion in mESC and mouse models is associated with a newly reconfigured α-globin self-interacting domain in which decreased interactions between the α-globin enhancers and the α−genes are replaced by newly formed interactions between the α-globin enhancers and the *Rhbdf1* and *Snrnp25* genes (Figs. 3b, 4E SE$^{INV}$-WT track). The newly formed α-globin enhancer-promoter interactions were validated by reciprocal capture from the promoters of all genes throughout the 165 kb α-globin TAD including the *Rhbdf1* and *Snrnp25* genes (Supplementary Fig. 5, SE$^{INV}$-WT track).

In addition to the changes in chromatin accessibility and expression at the α-globin promoters when the α-globin SE is inverted, we also noted changes in chromatin accessibility, chromatin modifications and expression at the *Rhbdf1* and *Snrnp25* genes. Normally these genes lie upstream (5') of the α-globin enhancers but in the inversion, they lie downstream of the SE. Whereas *Rhbdf1* and *Snrnp25* are normally silent or expressed at low levels in WT mice respectively, both are clearly upregulated (2500x and 12x respectively based on normalised RNA-seq read counts) by inverting the SE in definitive erythroid cells derived from the mouse SE$^{INV}$ model (Fig. 4b, d). Of particular interest *Rhbdf1* is normally repressed by the Polycomb system in definitive erythroid cells but in the SE$^{INV}$ erythroid cells, the associated chromatin modification (H3K27me3) is completely erased and H3K4me3 acquired as the gene is activated (Fig. 4a). We observe similar effects on *Rhbdf1* and *Snrnp25* in embryonic E10.5 and EB-derived primitive erythroid cells (Supplementary Fig. 3b, c). Inversion of the SE did not affect interactions or expression of any other genes lying up to 5.6 Mb either side of the 165 kb TAD (Supplementary Fig. 6a, b). Thus, inverting the α-globin SE within the 165 kb TAD redirects enhancer/promoter interactions causing decreased expression of the α−genes, which normally lie downstream, and activation of two genes (*Rhbdf1* and *Snrnp25*) normally lying upstream of the SE. These findings are consistent with the proposal that the cluster of enhancer-like elements within the SE work in an orientation-dependent manner within the 165 kb TAD.

### A CTCF insulator element does not explain the directionality of the inverted SE

We previously characterized two CTCF insulator elements (termed HS3839) which normally lie between the α-globin SE and the three genes (*Snrnp25, Rhbdf1* and *Mpg*) flanking the 5' boundary of the self-interacting domain[21]. Removal of this boundary leads to modest activation of the three genes flanking the α-globin sub-TAD but does not affect α-globin expression[21]. The inversion moves HS3839 from its native position, between the 5' flanking genes and the SE, such that it now lies between the enhancers, the *Mpg* gene, and the α-globin genes but in the opposite orientation (Fig. 1b i, v, Supplementary Fig. 1c). Chromatin accessibility and binding of CTCF to this re-positioned element appeared identical to that seen in its native position (Figs. 3a, 4a, Supplementary Fig. 3a). Therefore, rather than a change in orientation of the SE, we asked whether the observed changes in chromatin conformation and gene expression could simply result from changing the position and orientation of this boundary element.

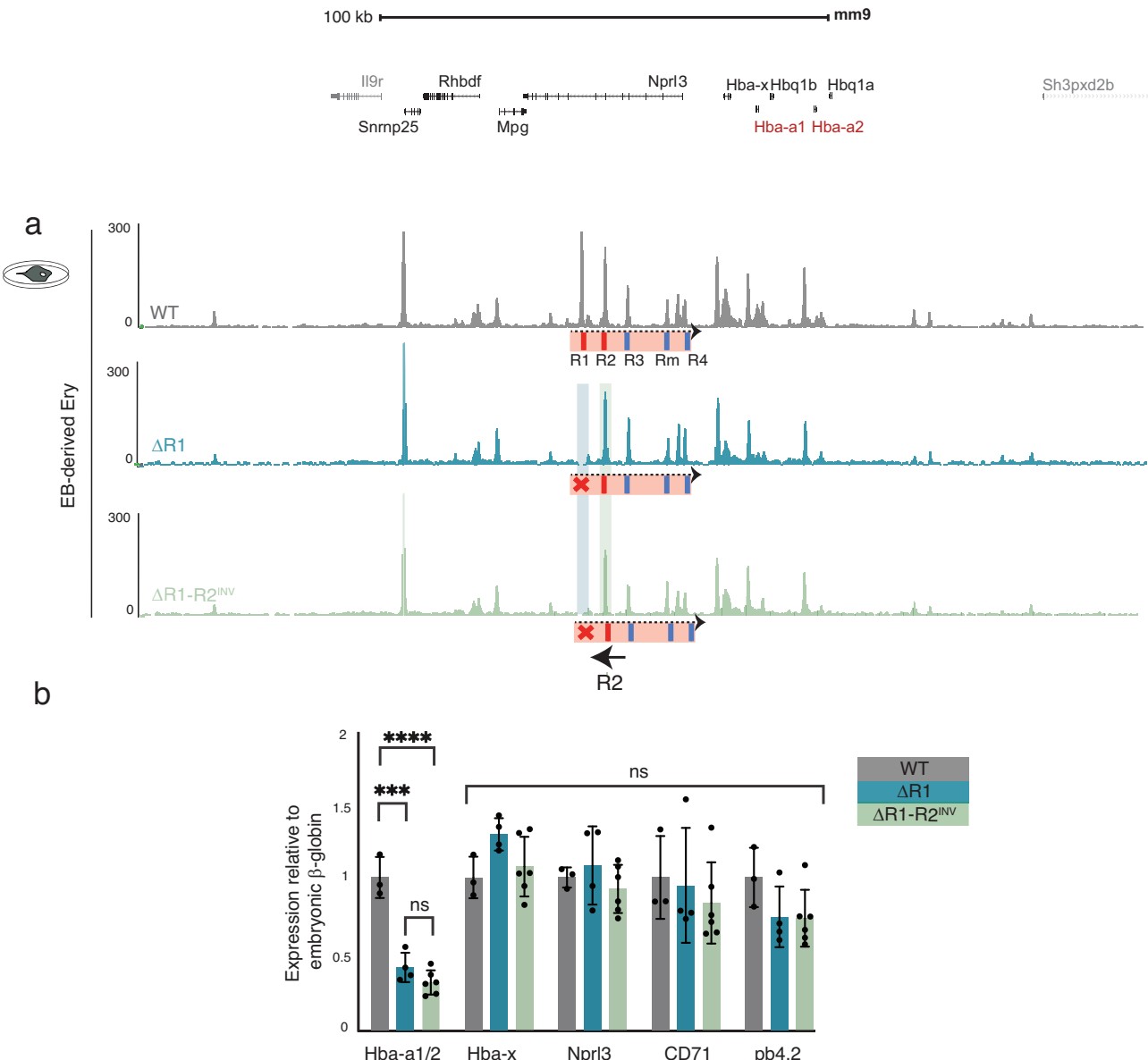

**Fig. 2 | The inversion of the major α-globin enhancer (R2) has no detectable effect on the locus. a** Top, RefSeq gene annotation. ATAC-seq tracks show chromatin accessibility profiles in erythroid cells derived from wildtype (WT, grey track) and R1 enhancer deletion (ΔR1) mESC models (blue track). Note the absence of the peak corresponding to R1 on the blue track, highlighted with a shaded blue bar. SE schematics as in Fig. 1 (i, ii). ATAC-seq track for the ΔR1-R2$^{INV}$ (green) indicates deletion of R1 (shaded blue bar) and intact open chromatin over the inverted enhancer R2 (shaded green bar). **b** Gene expression analysis by real-time qPCR assessing levels of mRNA for *Nprl3, CD71, pb4.2* as controls for the analysed erythroid population, as well as the α-globin genes (adult *Hba-a1/2 and* embryonic *Hba-x*), relative to the embryonic β-like globin gene (*Hbb-h1)* and normalised to WT. Independent erythroid differentiation experiments were analysed for each genetic model; biological replicates WT $n = 3$, ΔR1 $n = 4$, ΔR1-R2$^{INV}$ $n = 6$. Error bars indicate the standard deviation (SD) and black dots represent individual data points. Statistical analysis was performed using two-way ANOVA and Tukey post-hoc test: ****$p < 0.0001$, ***$p < 0.001$, ns: non-significant. Source data including statistical analysis is available in the Source Data file.

An initial observation argued against this. The engineered rearrangement induced here, which repositioned the insulator element and inverted the SE, had a much greater effect on interaction and expression of the *Rhbdf4* and *Snrnp25* genes than simply deleting the insulator element[21]. This suggests that the predominant effect on these 5' flanking genes resulted from inverting the SE rather than deleting the insulator element. To highlight the effect of the inverted SE without the confounding effect of the insulator elements, we evaluated the role of HS3839 in the normal locus versus the inverted locus by comparing erythroid cells harbouring HS3839 deletion in the natural configuration of the locus on both alleles (ΔCTCF/ΔCTCF), to those in which HS3839 has been removed from both inverted alleles (SE$^{INV}$-ΔCTCF / SE$^{INV}$-ΔCTCF), in EB-derived erythroid cells resulting from the in vitro

differentiation of the engineered models in mESCs (Fig. 1b iv, v, vi, Fig. 5a). We found that the major changes (chromatin state, expression and interactions) seen in the SE$^{INV}$ model persisted in the SE$^{INV}$-ΔCTCF model except for the *Mpg* gene (described in the section below). Therefore the reported SE$^{INV}$ changes were not completely confounded by an insulator effect (comparing data from (ΔCTCF to SE$^{INV}$-ΔCTCF, Fig. 5b, c, Supplementary Figs. 8, 9a) nor fully reversed by removing HS3839 from the inverted allele (comparing SE$^{INV}$ to SE$^{INV}$-ΔCTCF, Supplementary Fig. 7). In the absence of HS3839, the direction in which the SE exerts its effect appears to be partly encoded within the SE itself. In summary, re-positioning of the HS3839 element is not the only cause for reduced interaction between the SE and the α-globin genes nor it is solely responsible for the perturbed expression of the

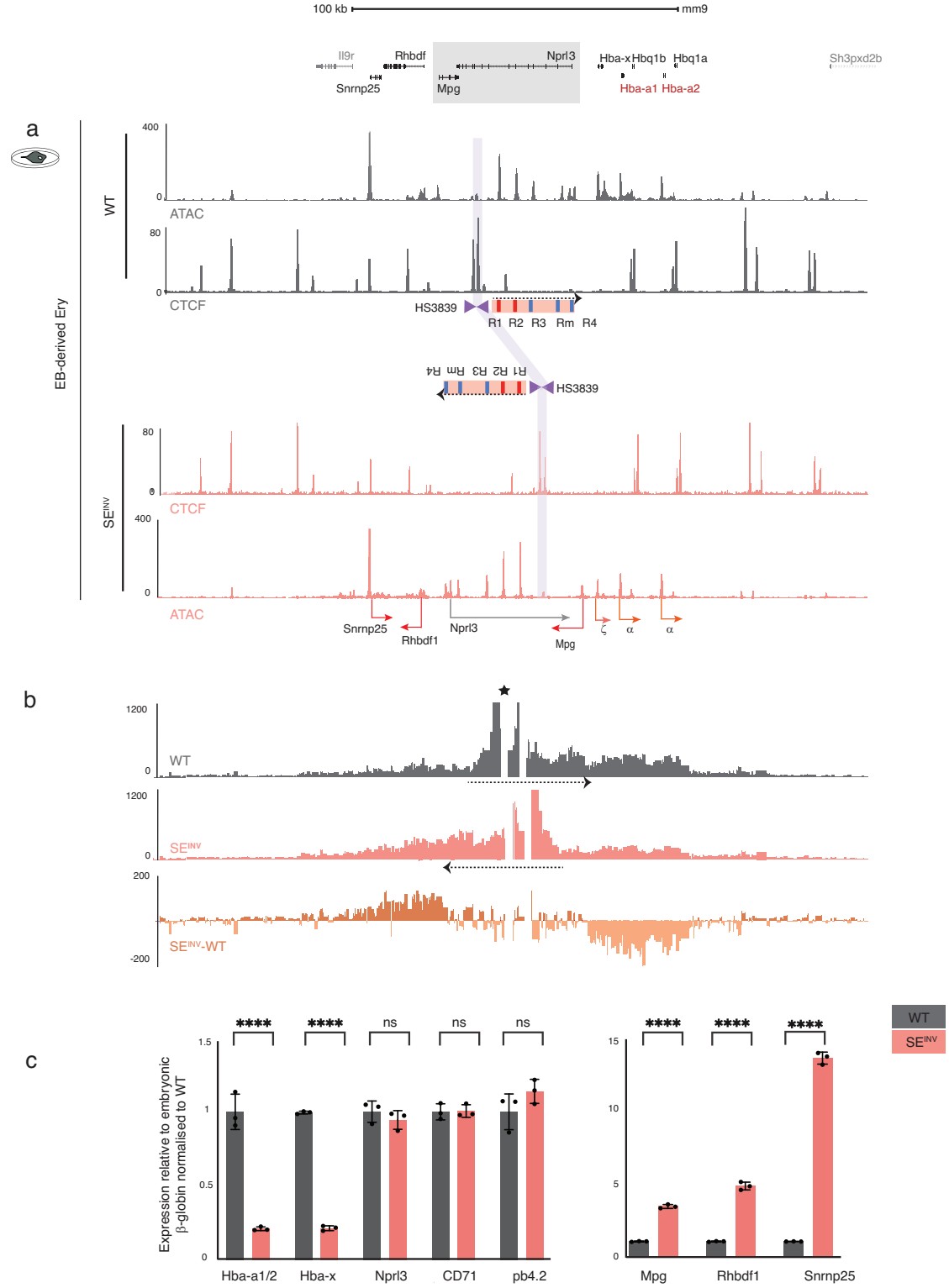

surrounding genes, including the upregulation of *Rhbdf1* and *Snrnp25* genes.

### The repositioned Mpg gene in inversion model has no impact on SE^INV phenotype

A prominent change in interaction we noted was between the SE and the repositioned *Mpg* promoter in the inverted allele without the boundary element (SE^INV-ΔCTCF) compared with the inversion alone (SE^INV) or the WT allele without the boundary element (ΔCTCF), (Fig. 5b, Supplementary Fig. 7). We also found that this was accompanied by a three-fold increase in expression of *Mpg* (Fig. 5c). In both inverted alleles (SE^INV and SE^INV-ΔCTCF), the *Mpg* promoter is located between the α-globin SE and the α-globin genes. We hypothesized that the *Mpg* promoter could act as a new insulator element as proposed

**Fig. 3 | Perturbed SE interactions and surrounding genes' expression profiles in SE$^{INV}$ EB-derived erythroid cells. a** Top, RefSeq gene annotation. Below, normalised (reads per kilobase per million mapped reads, RPKM) and averaged read-densities from 3 independent experiments of ATAC-seq and CTCF ChIP-seq show open chromatin and CTCF occupancy in EB-derived erythroid cells differentiated from WT and SE$^{INV}$ mESCs. For the annotated schematics, refer to Fig. 1i and v. The purple shaded bar indicates the position of CTCF boundary elements (HS3839) in WT- and SE$^{INV}$-derived erythroid cells. **b** NG Capture-C interaction profiles in WT (grey) and SE$^{INV}$ (orange) EB-derived erythroid cells show normalised and averaged interacting fragment count using a 6 kb window (n = 3 independent biological replicates). Additional track shows subtraction (SE$^{INV}$-WT) per *Dpn*II fragment of significantly interacting fragments using DESeq2 (p.adj<0.05) with light orange for

reduced interactions and dark orange for increased interactions in SE$^{INV}$-derived erythroid cells. The dashed black arrows indicate the direction of the SE in both WT and SE$^{INV}$ models. The star marks the viewpoint (the R1 enhancer) used in the NG Capture-C experiment. **c** Gene expression analysis by real-time qPCR assessing levels of mRNA for controls *Nprl3, CD71, pb4.2*, and *Mpg, Rhbdf1*, and *Snrnp25* genes, as well as the α-globin genes (*Hba-a1/2 and Hba-x*) relative to the embryonic β-like globin gene (*Hbb-h1*) and normalised to WT. Independent erythroid differentiation experiments were analysed for each genetic model; biological replicates n = 3. Error bars indicate the standard deviation (SD) and black dots represent individual data points. Statistical analysis was performed using two-way ANOVA and Sidak multiple comparisons test: ****p < 0.0001, ns: non-significant. Source data including statistical analysis is available in the Source Data file.

for other promoters[29–32]. If so, the *Mpg* promoter may act as an insulator and thereby play a role in reducing α-globin expression and perhaps increasing expression of *Rhbdf1* and *Snrnp25* (Fig. 5c). To address this, we deleted the *Mpg* promoter (ΔMpg) from all mESC models (WT, ΔCTCF, SE$^{INV}$, SE$^{INV}$-ΔCTCF, Fig. 1B i*, iv*, v*, vii) and evaluated the effect of this deletion on expression of the other genes in the landscape (*Rhbdf1, Snrnp25, Hba-x* and *Hba1/2*) and their interaction profiles. ATAC-seq DNA accessibility profiles confirmed the deletion of the *Mpg* promoter as well as the integrity of all other elements in the domain (Fig. 6a). No significant changes in gene expression or interaction profiles were observed when comparing ΔMpg models (Fig. 1b i*, iv*, v*, vii) with all the clones they derived from (Fig. 6b, c, Supplementary Fig. 9b). Deleting *Mpg* did not restore expression of the α-like globin genes to that observed in the ΔCTCF or WT models (Fig. 6c, Supplementary Fig. 9b). In summary, the repositioned *Mpg* gene makes no contribution to the change in the direction of interaction or activity of the SE.

### Cohesin, Med1 and PolII distribution reflects the direction of SE activity

Inversion of the SE at the α-globin locus in definitive erythroid cells reduces α-globin expression to half of its normal level and increases expression of the upstream genes by ~10 fold, overcoming Polycomb-mediated repression. This cannot be explained by many determinants of enhancer-promoter selectivity such as enhancer-promoter compatibility[33,34], distance of the SE to the genes involved[35], or the rearrangement of promoters or CTCF binding sites in the landscape[36,37]. We have recently shown that the level of cohesin within the α-globin sub-TAD is enriched ~6 fold in the presence of all α-globin enhancer-like elements compared to α-globin SE knockout models[10,38]. We have also reported that cohesin loop extrusion appears to be an important contributory mechanism in the regulation of globin gene expression[39]. Specifically, as in other cell-specific sub-TADs, cohesin appears to be recruited at enhancers from where it predominantly, but not exclusively, migrates to its cognate promoters[39]; another observation that points to a functional polarity in the regulatory landscape, driven by the enhancers. To examine how inversion of the SE may have impacted on cohesin distribution across the locus, we performed a Rad21 (one of the four core subunits of the cohesin complex) ChIP on erythroid cells derived from WT, SE$^{INV}$, and SE$^{INV}$-ΔCTCF-ΔMpg mESC models. We observe a pattern of Rad21 binding that mirrors the redirected functional activity of the SE; less cohesin-binding at the α-globin genes and more binding over *Rhbdf1* and *Snrnp25* gene loci (Fig. 7a, b, Supplementary Fig. 9c). This data corroborates our observation in the WT locus that the translocation of cohesin is directed by the SE, following the natural structure, from the enhancers R1-R2 towards the facilitator R4. Similar skewed distribution pattern was observed for PolII and the mediator complex component Med1 (Supplementary Fig. 10). This polarity does not seem to be affected by the presence of an insulation provided by the CTCF sites or a transcribing *Mpg* gene.

## Discussion

Here we have shown that, in contrast to individual enhancers, a SE containing multiple enhancer-like elements may act in an orientation-dependent manner. This adds to the increasing number of genomic features that may contribute to the specificity of enhancer-promoter interactions and the levels to which enhancers can increase gene expression. It is now clear from several experimental models that enhancers preferentially interact with promoters contained within a shared TAD[40–42] and that the interactions are limited by the linear distance between enhancers and promoters: the greater the distance the less the effect[35,43]. There is also evidence that some enhancers are more "compatible" with some promoters than others[34,44]. This suggests that some structural constraints exist either with respect to the associated chromatin or between multiprotein complexes at the enhancers and promoters. In some cases, enhancer-promoter interactions may depend on the recruitment of specific transcription factors or even complexes in a defined order implying structural constraints[45,46]. All of these features can combine to create a situation in which more than one promoter may compete for the activity of an enhancer.

It is also widely accepted that, in most cases, enhancers activate gene expression when they come into proximity to the promoter. Many mechanisms are proposed including the process of thermal diffusion[13,14,47] or more specifically cohesin-mediated chromatin loop extrusion[48–51]. We have shown that when the α-globin enhancers are active, peaks of the cohesin complex and its loader (Nipbl) appear at these elements suggesting that they could be the entry sites for increased cohesin loading in erythroid cells[21,38,52]. Similar conclusions have been drawn by others[43]. The translocation of cohesin can be interrupted by a variety of proteins including replication and transcription complexes[53–55] but, most notably, by appropriately orientated CTCF-bound insulator elements[36]. This in turn is associated with a reduction in enhancer-mediated gene expression. We have shown that the translocation of cohesin in the α-globin cluster and gene expression can be preferentially blocked by CTCF sites in which the N-terminus of bound CTCF is orientated towards the enhancer rather than the promoter[39]. This suggests that in the α-globin cluster, cohesin is loaded at the enhancer-like elements and predominantly, but not exclusively, translocates towards the promoters.

Together these observations raise the question of why inversion of the α-globin SE acts in an orientation-dependent manner. Importantly, in this inversion, the distance between the enhancers (R1, R2) and the promoters is changed by only 5 kb which, based on previous observations is very unlikely to lead to such a dramatic change in gene expression. The original inversion placed CTCF sites (HS3839) between the enhancers and promoters, but subsequent removal of these sequences clarified the distinct effects both the CTCF sites and the SE inversion had on the downregulation of α-globin expression. Similarly, the inversion placed a gene (*Mpg*) and its active promoter between the enhancers and the α-genes but again removal of this gene had little effect on α-gene expression. Importantly, the model in which

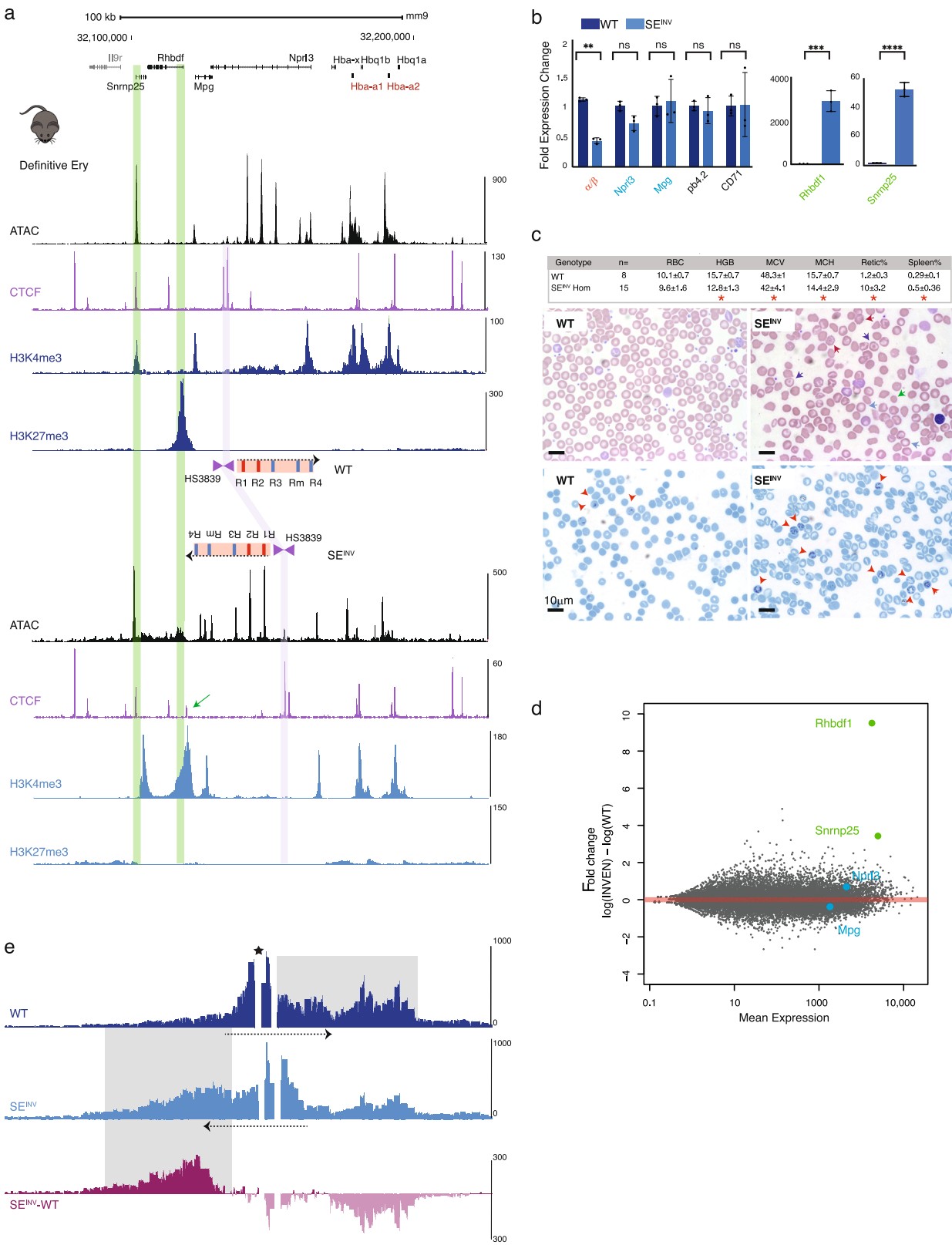

both the CTCF site and the *Mpg* gene were removed did not restore the α-globin expression and the accompanying chromatin state changes nor did it attenuate the upregulation of the genes located at the 5' of the α-locus. These observations suggest that the orientation of the composite SE itself plays a role in changing the direction in which it predominantly acts. Of interest, we have recently shown that the SE is an asymmetric structure consisting of two classical enhancers (R1 and

R2) and three facilitators (R3, Rm and R4) arranged as 5' R1, R2, R3, Rm and R4-3' with respect to the promoters[10]. It is possible that this either directs the predominant direction of loop extrusion towards the promoters or produces a structure that is more compatible with the promoters in its normal configuration (Fig. 8).

These findings are not explained by a model in which the enhancers simply form a transcriptional hub which favours

**Fig. 4 | SE$^{INV}$ perturbs gene expression and chromatin interactions at the α-globin locus and causes an α-thalassemia phenotype in a mouse model. a** Top, RefSeq gene annotation. Normalised (RPKM) and averaged read-densities of ATAC-seq, CTCF, H3K4me3 and H3K27me3 ChIP-seq ($n$ = 3) in Ter119+ spleen-derived definitive erythroid cells from both WT and SE$^{INV}$ mice. For the annotated schematics, refer to Fig. 2i and v. The purple shaded bar indicates the position of CTCF (HS3839) in WT- and SE$^{INV}$-derived erythroid cells. **b** Gene expression analysis as in Fig. 3c except: for the α-globin gene expression, the ratio of adult α-globin to adult β-globin is calculated (α/β). Three mice per genotype were analysed, $n$ = 3. **$p$ < 0.009, ns: non-significant. An unpaired $t$-test was performed on *Rhbdf1* and *Snrnp25* data: ***$p$ < 0.0005, ****$p$ < 0.0001. **c** Top, hematological parameters of red cells: Red Blood Cell (RBC) count, Hemoglobin measurement (HGB), Mean Corpuscular Volume (MCV, fL), Mean Corpuscular Hemoglobin (MCH, g dl⁻¹), the reticulocyte percentage (retic%), the spleen weight as a percentage of body weight (Spleen%) for WT and homozygous SE$^{INV}$ mice. Statistical analysis was performed using one-way ANOVA with a Tukey post-hoc test: red star for $p$ < 0.0001. Below, blood films (upper panels) and Brilliant Cresyl Blue (BCB)-stained blood (lower

panels) from WT and homozygous SE$^{INV}$ mice are shown. Abnormal red blood cells, characteristic of α-thalassemia, indicated by coloured arrows. Red: spiky cell membrane (acanthocytes), green: small and round cell (spherocyte), purple: target cells, blue: poorly hemoglobinized (hypochromic) cells, and red arrow-heads for immature erythroid cells (reticulocytes). **d** MA (log ratio (M) versus average (A)) plot of RNA-seq data derived from WT and SE$^{INV}$ primary definitive erythroid cells ($n$ = 3). Mean RNA abundance is plotted on the x-axis and enrichment is plotted on the y-axis. Green dots: significantly upregulated genes (*Rhbdf1*, *Snrnp25*) in the SE$^{INV}$ and blue dots for controls (*Nprl3* and *Mpg*), unaffected by the SE inversion. **e** NG Capture-C interaction profiles in WT (navy) and SE$^{INV}$ (blue) show means ($n$ = 3) of interacting fragment count using a 6 kb window. Additional track shows subtraction (SE$^{INV}$-WT) per *Dpn*II fragment of significantly interacting fragments using DESeq2 ($p$.adj<0.05); light pink for reduced interactions and dark pink for increased interactions in SE$^{INV}$-derived erythroid cells. Dashed black arrows indicate the SE direction. Star marks the viewpoint (the R1 enhancer). Source data including statistical analysis is available in the Source Data file.

transcription and to which promoters intermittently gain access. Such a model would not easily explain orientation dependence (Fig. 8). It might also be argued that the inversion has changed the "structure" of the locus and this, in turn, has altered the function. However, the structure of the locus is the product of the cis-elements and their interactions which simply reposes the question of why the orientation of the SE matters.

Finally, the question arises of whether the findings presented here are generally true of other SEs. Given that in the past, principles of gene expression established at the globin loci have always been found at the majority of mammalian genes rather than being specific to the globin genes, this seems likely. Of interest, in one set of experiments using large, randomly integrated transgenic inserts derived from bacterial artificial chromosomes, it was found that inversion of the β-globin LCR (a well-characterised SE) reduced expression of the linked β-globin gene cluster[56]. In addition, when the β-globin LCR was inserted in either orientation within a group of housekeeping genes, although it activated genes both 5' and 3', the upregulated genes changed depending on the orientation of the SE[57]. Finally, when the complex enhancer cluster lying between the *Kcnj2* gene and the *Sox9* gene was inverted, there was a reciprocal change in expression of these two genes[58]. Together, these observations suggest that, whereas single enhancers act in an orientation-independent manner, clusters of enhancers may act as a unit with an encoded bias to the direction in which they activate gene expression. However, it remains unclear whether such directionality is a general feature of SEs and what the underlying mechanisms might be. To examine this will require detailed genetic engineering of other SEs as set out here.

## Methods

### Mice

All mouse work was performed in accordance with UK Home Office regulations under project license number 30/3339 with the appropriate local ethical review completed by the University of Oxford Medical Sciences Division Animal Welfare and Ethical Review Board (AWERB). All animals were housed in Individually Ventilated Cages (IVC) with enrichment, provided with food and water ad libitum, and maintained on a 12 h light: 12 h dark cycle (150-200 lux cool white LED light, measured at the cage floor). Mice were given neutral identifiers and the haematological phenotype was analysed by research technicians unaware of mouse genotype during outcome assessment. Sex of the animals has no impact on the molecular analyses in this study and was not included as a factor except for the hematological assessment, where equal number of males and females was included and shown no impact on outcome. Samples for ATAC-seq, ChIP-seq, NG Capture-C, and gene expression analyses

were not randomized and the investigators were not blinded to allocation during these experiments and outcome assessments. The α-globin super-enhancer inversion (SE$^{INV}$) mouse model was generated by blastocyst injection of targeted E14-Tg2a mouse Embryonic Stem Cells (E14 mESCs, 129/Ola strain) heterozygous for SE$^{INV}$ into a C57BL/6 J mice. E14-Tg2a are deficient for *Hprt* expression resulting from deletion of part of the endogenous *Hprt* gene[59].

### Isolation of erythroid cells from adult mouse spleen

Primary Ter119+ erythroid cells were obtained from the spleens of adult mice that were treated with phenylhydrazine as previously described[60]. Spleens were mechanically dissociated into single cell suspensions in cold phosphate-buffered saline (PBS; Gibco: 10010023)/10% fetal bovine serum (FBS; Gibco: 10270106) and passed through a 70 μm filter to remove clumps. Cells were washed with cold PBS/10% FBS and resuspended in 10 μl of cold PBS/10% FBS per 10⁶ cells and stained with a 1/100 dilution of anti-Ter119-PE antibody (BD Pharmingen 553673 2 μg/ml) at 4 °C for 20 minutes. Stained cells were washed with cold PBS/10% FBS and resuspended in 8 μl of cold PBS/0.5% BSA/2 mM EDTA and 2 μl of anti-PE MACS microbeads (Miltenyi Biotec: 130-048-801) per 10⁶ cells and incubated at 4 °C for 15 minutes. Ter119+ cells were positively selected via MACS lineage selection (LS) columns (Miltenyi Biotec: 130-042-401) and processed for downstream applications. Purity of the isolated erythroid cells was routinely verified by Fluorescence-Activated Cell Sorting (FACS).

### Hematological analysis of the SE$^{INV}$ mouse model

Hematologic parameters for adult mice from SE$^{INV}$ model: Wild-type (WT), heterozygous (Het) and homozygous (hom) littermates were assessed where possible. A balanced mix of male and female were examined at more than 7 weeks of age to minimise between-individual variation. All mice were generated on the same complex background. Blood was collected into heparinized capillary tubes and blood counts were assayed using the Horiba Medical SciI Vet abc Plus+ instrument. Sample sizes were calculated based on previous observations where a SD of 7% is a realistic estimate of the variability. Using this figure, in conjunction with the 95% confidence interval and a power of 0.8 (a standard figure) to detect a true positive, we needed a sample size of 7 mice to detect a 10% change in the above parameters using a 2-sided statistical test. Blood smears were prepared using 2 μl of peripheral blood spread on a slide, dried, stained, and mounted. The blood smears were either used for the reticulocyte counts performed manually after Brilliant Cresyl Blue (BCB) staining or for general red blood cell morphology assessment after May–Grünwald–Giemsa (MGG) staining. Reticulocyte counts and blood smear assessment were performed by two independent assessors blinded to genotype.

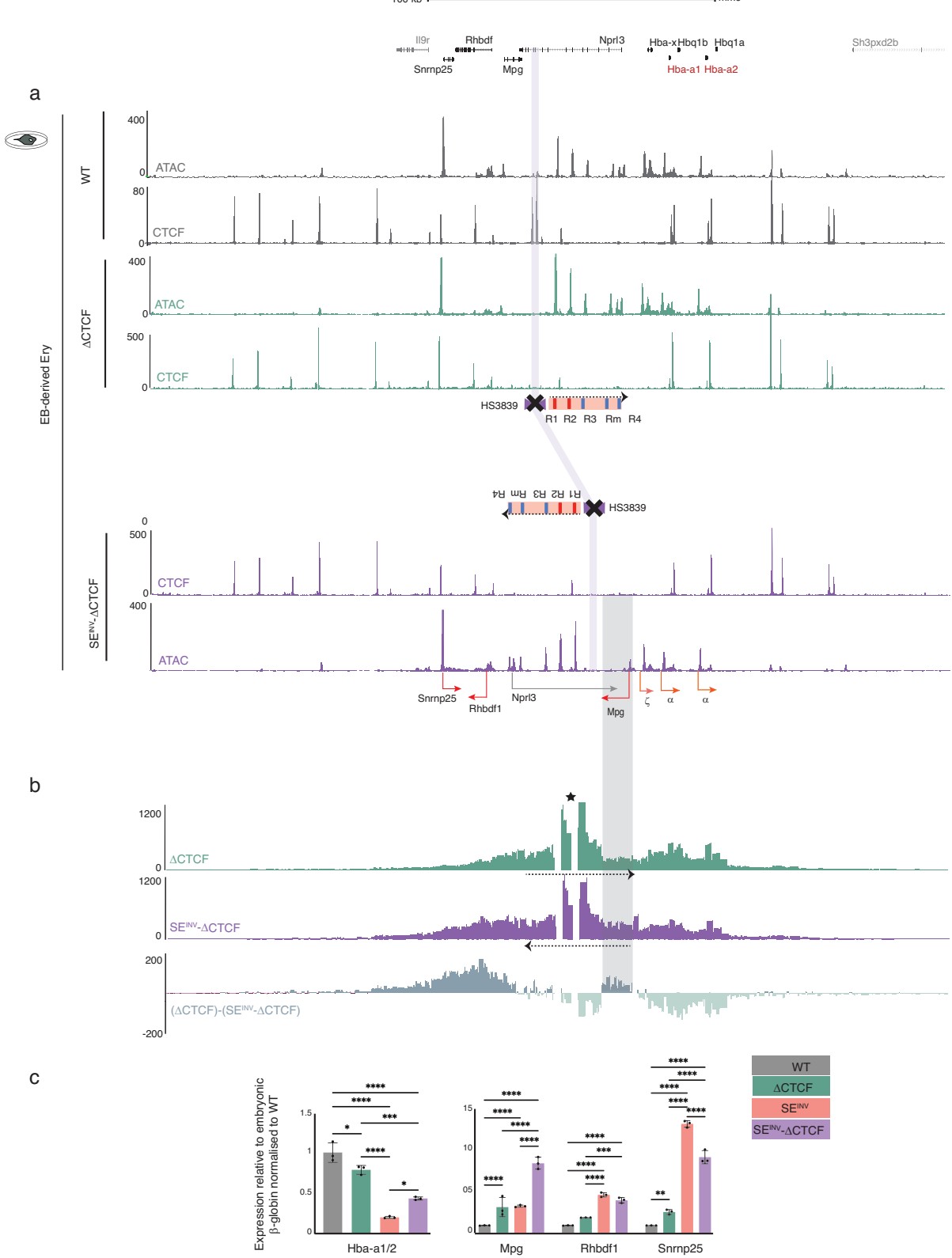

**Mouse Embryonic Stem Cells (mESCs) targeting strategy to generate the inversion of the α-globin SE (SE^INV model)**

The inverted region encompasses the α-globin SE but was delimited such that the breakpoints do not impinge on the integrity of the locus including the transcription of *Nprl3* and *Mpg* housekeeping genes. The strategy also ensured that the chromatin at the boundaries of the inversion are not marked by any histone modifications or bound by

any of the factors assessed (Supplementary Fig. 1b), to avoid disruption of a potentially functional DNA regions. The genome engineering was based on a strategy[59] which involved sequential rounds of transfection and clone selection in mESCs, first to define the inversion boundaries by targeted integration of loxP sites, and then to create the inversion between these sites by Cre-mediated recombination. Cre-mediated recombination results in concomitant reconstruction of a

**Fig. 5 | Deletion of the 5′ α-globin CTCF boundary element (HS3839) in SE^INV mESC model does not rescue the SE^INV phenotype in in vitro EB-derived erythroid cells. a** Top, RefSeq gene annotation. Normalised (reads per kilobase per million mapped reads, RPKM) and averaged read-densities (n = 3) of ATAC-seq and CTCF ChIP-seq in EB-derived erythroid cells differentiated from WT, ΔCTCF and SE^INV-ΔCTCF mESCs. For the annotated schematics, refer to Fig. 1i, iv, and vi. The purple shaded bar indicates the position of CTCF boundary element (HS3839) in WT, ΔCTCF and SE^INV-ΔCTCF erythroid cells. **b** NG Capture-C profiles in ΔCTCF (green) and SE^INV-ΔCTCF (purple) show means (n = 3) of interacting fragment as in Fig. 4e. Subtraction track shows light green for reduced interactions and dark green for increased interactions in SE^INV-ΔCTCF erythroid cells. The grey shaded area,

encompassing the *Mpg* gene located between the SE and the α-globin genes in the SE^INV models, indicates the increased interactions at the newly positioned *Mpg* in the SE^INV-ΔCTCF. **c** Gene expression analysis by real-time qPCR assessing levels of mRNA for controls *Nprl3*, *CD71*, *pb4.2*, as well as *Mpg*, *Rhbdf1*, and *Snrnp25*, and the α-globin (*Hba-a1/2* and *Hba-x*) relative to the embryonic β-like globin gene (*Hbb-h1*) and normalised to WT. Independent erythroid differentiation experiments were analysed for each genetic model; biological replicates n = 3. Error bars indicate the standard deviation (SD) and black dots represent individual data points. Statistical analysis was performed using two-way ANOVA and Sidak multiple comparisons test: ****$p < 0.0001$, ***$p < 0.001$, **$p < 0.005$, *$p < 0.05$. Source data including statistical analysis is available in the Source Data file.

functional hypoxanthine phosphoribosyl transferase (*Hprt*) mini-gene from a pair of defective *Hprt* genes, each part integrated in one of the two the targeting vectors, at the 3′ and 5′ of the inversion. Similar approach was followed previously at the mouse α-globin locus[61]. In outline, two targeting vectors, each harbouring a LoxP site flanked by sequences homologous to either the 5′ or the 3′ insertion sites at the α-globin locus and one part of a the *Hprt* mini-gene, were assembled following a recombineering strategy in *E. Coli* (Supplementary Fig. 11a). To allow for assembly of the targeting vectors in bacteria, the selection cassettes used in the 5′ and 3′ targeting vectors were modified to be driven by a prokaryotic promoter. After assembly in BACs and retrieval onto a plasmid (by recombineering), the targeting vectors were electroporated into mESCs for a sequential insertion of the 3′ and then 5′ convergent LoxP sites at the α-globin locus (insertion sites at the edges of this interval, chr11:32,224,538-32,271,838). Positive resistance markers in the 3′ and 5′ targeting constructs (Hygromycin and Neomycin respectively) secured the selection for LoxP integration along with a defective 5′ or 3′ components of an *Hprt* mini-gene at the flanks of the ~48Kb sequence. Sequential employment of site-specific recombinases induces the inversion of the 48Kb sequence (Cre), and deletion of all selectable markers (Flp) (Supplementary Fig. 11b, c). Upon inversion induced by Cre, selection is done by hypoxanthine, aminopterin, thymidine (HAT)-resistance where only in mESCs where the integration of both targeting vectors occurred *in cis* and resulted in the reconstruction of the *Hprt* gene (3′ and 5′ parts of the gene) upon successful inversion would survive. Any other events would result in nonviable mESCs (dicentric chromosomes). 6-thioguanine (6-TG) is used to select for the deletion of the selection markers upon Flp. Southern blots and PCR strategies were used for the screening for targeted events both at the 3′ and 5′ integration sites (Supplementary Data 1). Supplementary Table 1 contains primers for the Southern blot probe amplicons. The efficiency of targeting was ~39% for the 3′ LoxP site integration (60 positive clones out of 153 screened). The efficiency of targeting at the 5′ end was 53% (138 positive clones out of 259 screened). The inversion was detected and confirmed by Southern Blot and PCR at 68% (69 clones out of 96 screened). Flp excision was 100%.

To generate homozygous SE^INV/SE^INV mESCs, we retargeted clones heterozygous for SE^INV (WT/ SE^INV) using two selection rounds; 5′ Neomycin cassette to introduce the 5′ LoxP site and HAT to select for the simultaneous insertion of the 3′ LoxP site and Cre-recombination which reconstructs the *Hprt* gene. The strategy and data for retargeting clones heterozygous for SE^INV(WT/ SE^INV) following *Hprt* mini-gene excision in the resulting homozygous SE^INV/SE^INV clones (A4.2, B4.4, F11.2) is shown in (Supplementary Fig. 12a-d). The integrity of the SE^INV homozygous clone (A4.2) was checked using an orthogonal method; optical mapping using Bionano Saphyr technology.

### Optical mapping using Bionano Saphyr technology
High molecular weight DNA from mESC (wild-type E14 and genetically engineered homozygous for the inversion A4.2 SE^INV/SE^INV) was isolated using the Bionano plug lysis protocol, following the Bionano Prep Blood and Cell Culture DNA isolation kit. Genomic DNA was barcoded

using the Bionano Prep DLS labelling kit, in accordance with the manufacturer's instructions. Labelled samples were analysed using the Saphyr device, and optical maps were assembled and structural variants analysis was conducted using Bionano Access software.

### CRISPR-Cas9 strategies
For generation of enhancer R1 deletion (ΔR1) and R2 inversion (R2^INV) mESCs, the same strategy was followed; two sgRNAs that flank the enhancer sequence were designed and cloned into a pSpCas9(BB)-2A-GFP (pX458) vector, a gift from Feng Zhang (Addgene plasmid #48138; http://n2t.net/addgene:48138; RRID:Addgene_48138) or a modified vector with the GFP tag exchanged for an mRuby cassette (pX458-Ruby). For generation of CTCF sites (HS3839) deletion (ΔCTCF) mESCs, sgRNAs were designed to flank a sequence encompassing the two CTCF sites (at HS-38: TACCCTCTGGTGGC and at HS-39.5: CTGGCCACTGGGGG separated by around 1.5 kb). Four guides for a CRISPR/Cas9 nickase strategy were cloned into p335 (Cas9-D10A) modified vector containing a neomycin selectable marker. An HDR vector of 2991 bp was designed such that a fragment encompassing both CTCF sites and the intervening 1447 bp sequence was flanked by homology arms of 750 bp. sgRNA recognition sequences outside the homology arms were also included and resulted in the release of the HDR sequence from the donor vector upon co-transfection with sgRNA/Cas9 plasmids. Protospacer Adjacent Motif (PAM) sequences were mutated where necessary to avoid cutting the HDR plasmid. CTCF consensus sequences were replaced by ScaI sites (AGTACT) for ease of screening. Modified sequences were checked using Sasquatch tools for undesirable creation of potentially active hypersensitive sites in erythroid cells[62]. Similar strategy was followed to produce ΔMpg except that HDR was achieved using a ssODN of 200 bp with 97 and 94 bp homology with the flanks of *Mpg* exon1 and a ScaI restriction site (bold) inserted in the middle for screening purposes. ssODN:TCTGTGGCCCACCCTATCCCAAGGCAGAGCCTTCTGGTCCGGG ACCCTTGAAAATTCAGTCCGGGTTCAGAGAGGATGTCTTGATGGCAAT CGGGCC**AGTACTT**GGACGCAGGAAAATCAGGAGTTCAAAGCCAGCATA GGCCACATGGGACCCTGTCCTTTAAAAGAGTAATGGCTGTTTAACTAG TCAGCTCAGC

### mESC transfections
To generate ΔR1, R2^INV, and ΔCTCF, mESC transfections were performed using the Neon electroporation system (Invitrogen), according to the manufacturer's instructions. A transfection protocol optimized for mESCs (3 pulses of 1400 V/10 ms) whereby 10^6 mESCs were resuspended in buffer R and electroporated with a total of 5 μg plasmid DNA (2.5 μg/gRNA for ΔR1 mESCs,) and 11 μg (1.25 μg/gRNA and 6 μg for HDR vectors for R2^INV and ΔCTCF). After electroporation, ΔR1 and R2^INV targeted cells were cultured for 48–72 hours prior to FACS sorting the eGFP-Ruby expressing population, which was then seeded at clonal density on gelatinized 10 cm dishes for at least 6 days. ΔCTCF cells were directly plated on a gelatinized 10 cm dish at clonal density and targeted clones were selected for neomycin resistance (G418 at 200 μg/ml) 24 h to 72 h post transfection and left to grow for at least 6 days. Colonies were picked into 96-well plates and screened for mutations by PCR and Sanger sequencing. For ΔMpg generation,

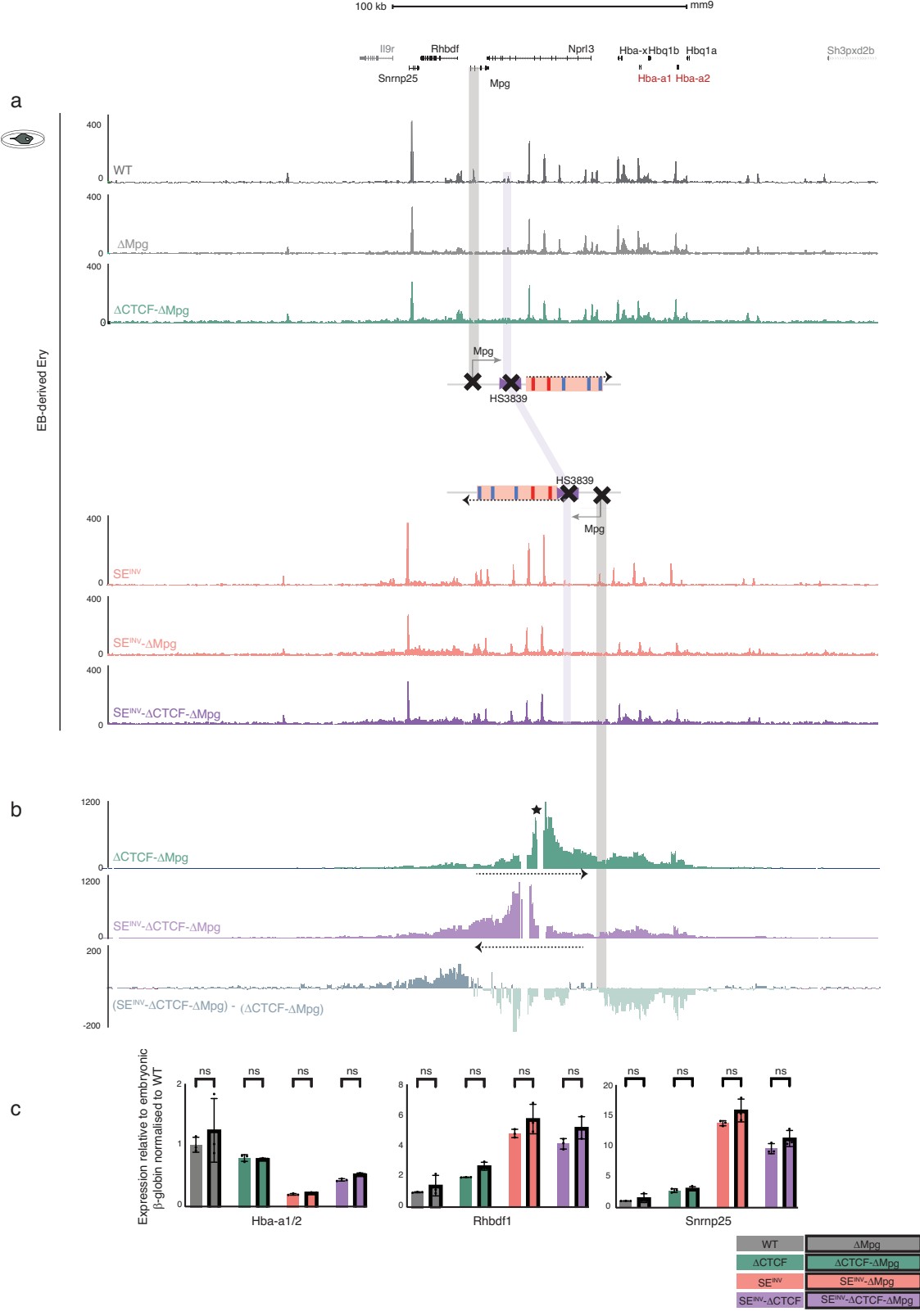

mESCs were transfected using the Fugene 6 Transfection Reagent (Promega), according to the manufacturer's instructions. In brief, 4 μg of total DNA (1 μg for each of sgRNA and 2 μg of ssODN HDR donor) were incubated with 20 μl Fugene 6 reagent/180 μl media for about 15 min before being added to 10⁶ of mESCs in a well of a 6-well plate. After 48hrs, the mESCs were FACS sorted for GFP and Ruby, the

reporters of the sgRNA vectors. Sorted cells were plated at clonal density (2000 cells/10 cm dish) and colonies picked (2 × 96 well plates/ mESC clone transfected) and screened using a restriction digest insertion; PCR the fragment across the expected deletion/insertion followed by ScaI digestion. For sgRNA and screening PCR primer sequences, see Supplementary Tables 2, 3.

**Fig. 6 | Deletion of the Mpg gene in SE[INV] and ΔCTCF mESC models does not rescue the SE[INV] model phenotypes in in vitro EB-derived erythroid cell. a** Top, RefSeq gene annotation. Normalised (RPKM) and averaged read-densities ($n = 3$) of ATAC-seq and CTCF ChIP-seq in EB-derived erythroid cells differentiated from mESCs models with native SE configuration and deletion of CTCF or/and Mpg promoter: WT, ΔMpg, ΔCTCF-ΔMpg, and mESCs models with inverted SE combined with the deletions of CTCF and Mpg: SE[INV], SE[INV]-ΔMpg, SE[INV]-ΔCTCF, and SE[INV]-ΔCTCF-ΔMpg. For the annotated schematics, refer to Fig. 1i and v. The purple shaded bar indicates the position of CTCF boundary element (HS3839) in WT- and SE[INV]-derived erythroid cells. **b** NG Capture-C interaction profiles in ΔCTCF-ΔMpg (green) and SE[INV]-ΔCTCF-ΔMpg (purple) show means of interacting fragment

counts ($n = 3$) as in Fig. 4e. Subtraction track shows reduced and increased interactions (light and dark green respectively) in SE[INV]-ΔCTCF-ΔMpg erythroid cells. The grey shaded bar indicates *Mpg* gene, located between the SE and the α-globin genes in the SE[INV] models. **c** Gene expression analysis for the α-globin gene (*Hba-a1/2*) as well as *Rhbdf1*, and *Snrnp25* relative to the embryonic β-like globin gene (*Hbb-h1*) and normalised to WT. Independent erythroid differentiation experiments were analysed for each genetic model; biological replicates $n = 3$. Error bars indicate the standard deviation (SD) and black dots represent individual data points. Statistical analysis was performed using two-way ANOVA and Sidak multiple comparisons test: ns: non-significant. Source data including statistical analysis is available in the Source Data file.

## mESC in vitro hematopoietic differentiation

The protocol is described in Francis et al.[22] 24–48 h prior to differentiation, cells were induced by passaging into base media (Iscove's modified Dulbecco's medium (IMDM), $1.4 \times 10^{-4}$ M monothioglycerol (Sigma-Aldrich) and 50 U/ml penicillin-streptomycin (Thermo Fisher)) supplemented with 15% heat-inactivated fetal calf serum (ΔFCS, Thermo Fisher) and 1000 U/ml LIF. For embryoid body (EB) generation, cells were disaggregated by trypsinisation and quenched in base media (as above) supplemented with 10% ΔFCS. Differentiation media was prepared fresh on the day of differentiation by supplementing base media (as above) with 15% ΔFCS, 5% protein-free hybridoma medium (PFHM-II, Thermo Fisher), 2 mM L-glutamine (Thermo Fisher), 50 μg/ml L-ascorbic acid (Sigma Aldrich), $3 \times 10^{-4}$ M monothioglycerol and 300 μg/ml human transferrin (Sigma Aldrich). Cells were plated in petri dishes (Thermo Fisher) at $1.5–3 \times 10^4$ cells in 10 ml differentiation media. EBs were left to differentiate for up to seven days without disruption except for gentle manual shaking every few days to disrupt potential EB aggregation and/or attachment to the dishes. EBs from 10 cm dishes were harvested by collection of the entire plate contents into falcon tubes, spinning at 300x g for 5 minutes before and after a single PBS wash step to ensure complete recovery. After PBS removal, EBs were disaggregated in 0.25% trypsin (0.3 ml per 10 cm dish) for 3 minutes at 37 °C with continuous manual shaking to prevent sedimentation at the bottom of the tube. Trypsin was quenched with an equal volume of FCS and a single-cell suspension obtained through tituration. Cells were collected by spinning at 300x g for 5 minutes and resuspended as needed.

## Erythroid population isolation from EBs

CD71 + (high) cells were isolated from disaggregated EBs by magnetic column separation (LS Column, Miltenyi), according to the manufacturer's instructions as previously described[22]. Briefly, cells were labelled with anti-mouse CD71-FITC (eBioscience 11-0711-85; 1:200) in staining buffer (PBS with 10% FCS; 500 μl per $10^7$ cells) for 20 minutes at 4 °C, rolling, then washed by adding staining buffer (1 ml per $10^7$ cells) and spinning. After supernatant removal, cells were incubated with MACS anti-FITC separation microbeads (Miltenyi; 10 μl per $10^7$ cells) in ice-cold separation buffer (PBS plus 0.5% bovine serum albumin (BSA) and 2 mM EDTA; 90 μl per $10^7$ cells) for 15 minutes at 4 °C, rolling, and washed by adding separation buffer (1 ml per $10^7$ cells) and spinning. Bead-labelled cells were resuspended in 500 μl cold separation buffer and added to a pre-equilibrated LS column. The negative fraction was washed through with two flushes of 3 ml cold separation buffer and the positive fraction collected by forcing cells from the column in 5 ml separation buffer. After spinning and supernatant removal, cells were resuspended in staining buffer as needed for downstream processing. Population purity and selection efficiency were determined by flow cytometry. Information on the antibodies used are in Supplementary Table 4.

## ATAC-seq and ChIP-seq

Assay for Transposase-Accessible Chromatin (ATAC)-seq was performed on 75,000 Ter119+ cells isolated from phenylhydrazine-treated

mouse spleens as previously described[19,63]. ATAC-seq libraries were sequenced on the Illumina Nextseq platform using a 75-cycle paired-end kit (NextSeq 500/550 High Output Kit v2.5: 20024906).

CTCF, H3K4me1, H3K4me3, and H3K27me3 Chromatin immunoprecipitation (ChIP) was performed on 5 to $10 \times 10^6$ Ter119+ erythroid cells using a ChIP Assay Kit (Millipore: 17-295) according to the manufacturer's instructions. Cells were crosslinked by a single 10 min 1% formaldehyde fixation. Chromatin fragmentation was performed with the Bioruptor Pico sonicator (Diagenode) for a total sonication time of 4 min (8 cycles) at 4 °C to obtain an average fragment size between 200 and 400 bps. Immunoprecipitation was performed overnight at 4 °C with various antibodies. Details of all antibodies for the chromatin immunoprecipitation experiments are in Supplementary Table 5.

For Rad21 ChIP, cells were double cross-linked using disuccinimidyl glutarate (DSG, Sigma) and 1% formaldehyde (Sigma) for a total fixation time of 1 hour. Fixed chromatin samples were fragmented using the Covaris sonicator (ME220) for 10 minutes (75 power, 1000 cycles per burst, 25% duty factor) at 4 °C. 50 μl per sample was removed as an input control. Sonicated samples were pre-cleared to remove background signal through incubation with a 1:1 mix of Protein A/G Dynabeads (InVitrogen). Antibody was added at a concentration of 5 μg/ml per sample and immunoprecipitated at 4 °C overnight (Supplementary Table 5). Immunoprecipitated samples were incubated with a 1:1 mix of Protein A/G Dynabeads for 5 hours at 4 °C. Beads were then washed 4x in RIPA buffer on a magnetic stand, followed by 1x wash with TE (Sigma Aldridge) + 50 mM NaCl (ThermoFisher). Chromatin was eluted from the beads using elution buffer and incubated for 30 minutes at 65 °C with shaking. Input samples were diluted 1:1 with elution buffer. Samples and input controls were incubated at 65 °C overnight for de-crosslinking, before RNase (Roche) and proteinase K (BioLabs) treatment. DNA fragments were purified using the Zymo ChIP DNA Clean & Concentrator kit (Zymo Research) and eluted in water. DNA concentration was quantified using the Qubit dsDNA HS assay (Invitrogen) as per manufacturers' protocol. To assess sonication efficiency the D1000 Tapestation (Agilent) assay was performed on an input sample. An approximately equal mass of input and IP DNA was used for indexing ($0.5 \text{ ng}^{-1}$ μg).

For Rad21 and Med1 ChIP-seq, $5 \times 10^6$ CD71+ cells were fixed with DGS (2 μM) for 30 minutes at room temperature, spun down and resuspended in 1% formaldehyde for 30 minutes at room temperature, rinsed in PBS and snap frozen prior to precipitation. Cells were lysed in SDS lysis buffer (1% SDS, 10 mM EDTA, 50 mM Tris-HCl ph8, 1x PIC) and sonicated to ~200-500 bp on a Covaris sonicator with the following settings: 600secs; 75 power; 25% Duty Factor; 1000 cycles per burst. Chromatin was diluted in dilution buffer (0.01% SDS, 1.10% Triton X-100, 1.2 mM EDTA, 16.7 mM TrisHCl ph8, 167 mM NaCl, 1x PIC). A 1:1 mix of proteinA:proteinG conjugated Dynabeads were added to chromatin prior to immunoprecipitation to reduce non-specific binding. Cleared chromatin was incubated with antibody with rotating overnight at 4 °C. Chromatin was then incubated with rinsed A:G dynabeads for 5 hours, to bind to the antibody. Beads were then washed 3x with RIPA buffer (50 mM Hepes-KOH, 500 mM LiCl, 1 mM EDTA, 1% IGEPAL CA-630, 0.7% Na-Deoxycholate) and chromatin was

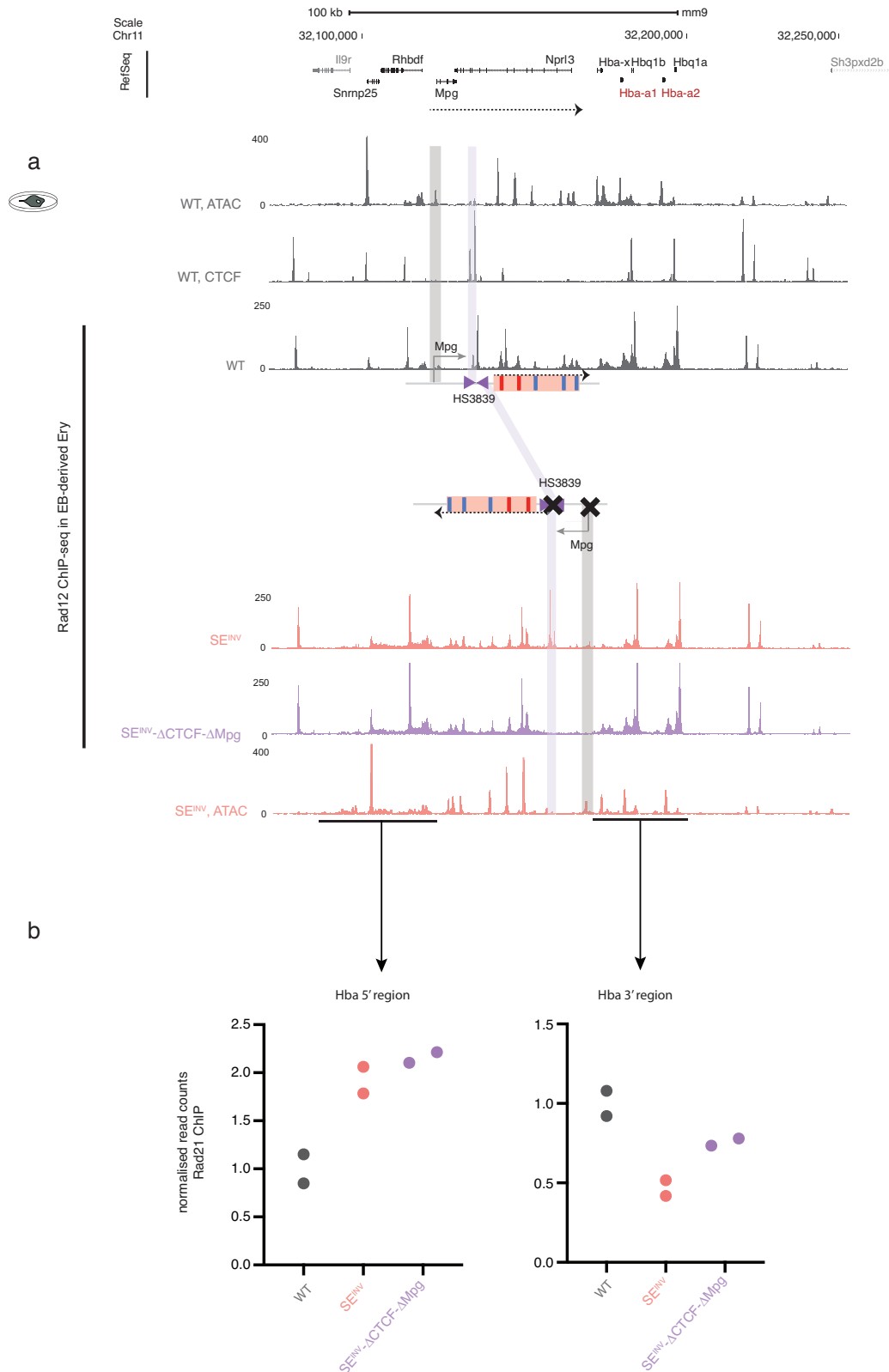

**Fig. 7 | Rad21 ChIP at the α-globin locus in erythroid cells derived from WT, SE$^{INV}$, and SE$^{INV}$-ΔCTCF-ΔMpg mESC models shows a cohesin distribution pattern that mirrors the SE sequence and functional orientation. a** RPKM-normalised ATAC-seq and CTCF ChIP-seq tracks for WT EB-derived erythroid cells for orientation. Below, representative RPKM-normalised Rad21 ChIP-seq tracks for WT, SE$^{INV}$ and SE$^{INV}$-ΔCTCF-ΔMpg EB-derived erythroid cells. ATAC-seq track for SE$^{INV}$ EB-derived erythroid cells for orientation. The schematic represents the WT (top) versus the SE$^{INV}$-ΔCTCF-ΔMpg (below) configurations. The grey and

purple highlighted areas indicate the *Mpg* gene and HS3839 CTCF sites respectively. **b** Rad21 ChIP read counts in two independent populations of erythroid cells derived each from WT, SE$^{INV}$, and SE$^{INV}$-ΔCTCF-ΔMpg in vitro differentiated mESCs, $n = 2$. Data shown for the regions 5′ and 3′ of the inverted SE and marked by a black line below the SE$^{INV}$ ATAC-seq track. The plots show the two replicates per genotype represented in two same-coloured dots. Reads are normalised to the average number of reads over a selected region in the β-globin locus.

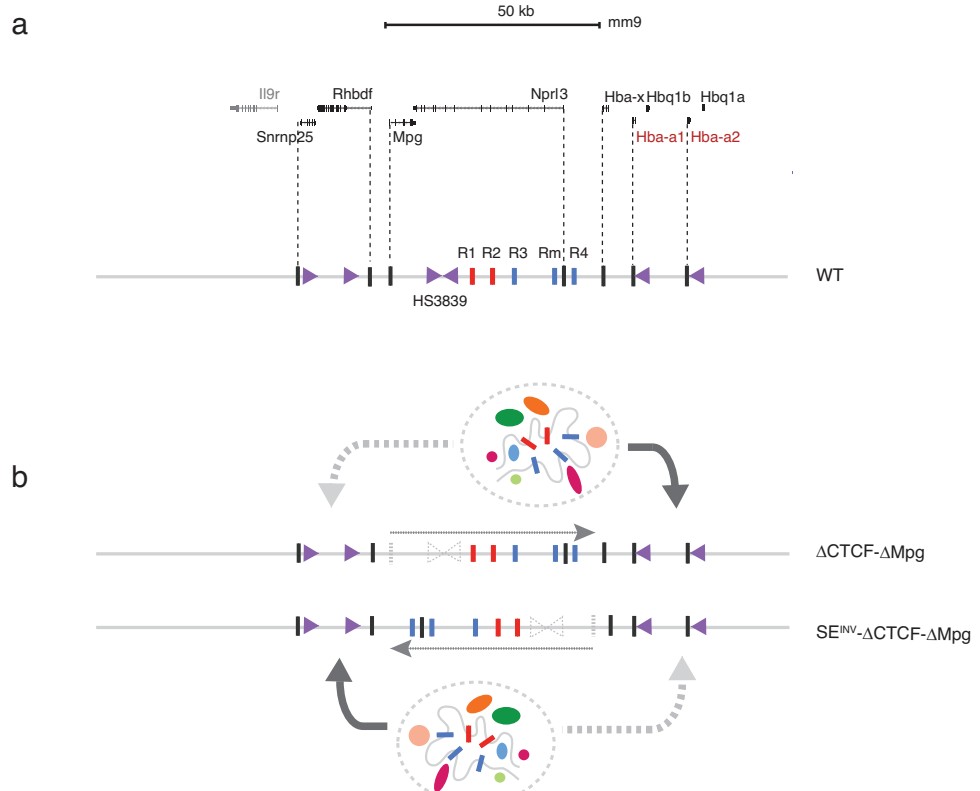

**Fig. 8 | A model based on a comparison of the engineered ΔCTCF-ΔMpg and SE^INV-ΔCTCF-ΔMpg α-globin loci. a** The α-globin locus as presented in RefSeq. Below, a schematic representation highlighting the key elements within the cluster including enhancers (red bars), facilitators (blue bars), and promoters (black bars). CTCF-bound insulators are represented by purple arrow heads pointing in the direction representing the orientation of the N-terminal domain of CTCF. HS3839 is the functional 5′ α-globin sub-TAD boundary. **b** The two engineered loci in which both the HS3839 insulators and the Mpg gene have been removed from the wild-type locus (ΔCTCF-ΔMpg) and the locus in which the SE has been inverted (SE^INV-ΔCTCF-ΔMpg). In both configurations, the SE would be expected to form a transcriptional hub: grey dashed ovals containing a graphical representation of the enhancer-like elements making contact and relevant transcription factors and co-factors are enriched (coloured varying size ovals). Nevertheless, the pattern of gene expression in the associated sub-TAD is still determined by the orientation of the SE (bold grey curved arrows). Weaker expression and interactions are represented by the light grey bold curved arrow. The predominant direction of chromatin interactions, determined by NG Capture-C and associated enrichment of cohesin (Rad21) is shown by dashed line arrows and follows the SE orientation.

eluted from beads using elution buffer (1% SDS, 10 mM EDTA, 50 mM TrisHCl ph8) and de-crosslinked at 65 °C overnight. Chromatin was treated with 0.5 μg RNase (60 minutes at 37 °C) followed by 20 μg Proteinase K (60 minutes at 37 °C). Finally, DNA was purified using the ChIP DNA clean and concentration kit (Zymo). DNA concentration was quantified using the Qubit dsDNA HS assay (Invitrogen) as per manu-facturers protocol. To assess sonication efficiency the D1000 Tapes-tation (Agilent) assay was performed on input sample. An approximately equal mass of input and IP DNA was used for indexing (0.5 ng–1 μg).

For PolII, we did a ChIPmentation. ~250,000 cells CD71+ cells were fixed with DGS (2 μM) for 30 minutes at room temperature, spun down and resuspended in 1% formaldehyde for 30 minutes at room temperature, rinsed in PBS and snap frozen prior to pre-cipitation. Cells were incubated on ice in lysis buffer (0.5% SDS, 10 mM EDTA, 50 mM Tris-HCl ph8, 1x PIC) and sonicated to ~200-500 bp on a Covaris ME220 sonicator with the following settings: 900secs; 75 power; 12% Duty Factor; 1000 cycles per burst. SDS was neutralized with triton X to a final concentration of 1%. A 1:1 mix of proteinA:proteinG conjugated Dynabeads in 0.5% BSA, 1x PIC were incubated with PolII antibody for 2–4 hours then washed twice with 0.5% BSA, 1x PIC. Sonicated chromatin equivalent to 250,000 cells was added to antibody-bound beads and incubated rotating over-night at 4 °C. Beads were then washed 3x with RIPA buffer (50 mM Hepes-KOH, 500 mM LiCl, 1 mM EDTA, 1% IGEPAL CA-630, 0.7% Na-Deoxycholate), 1x with 1X TE, and twice with 10 mM Tris (pH 8).

Beads were then incubated for 5 mins at 37 °C with 30 μl 1x tag-mentation buffer and 1 μl of Tn5 enzyme, then suspended in RIPA buffer to stop the reaction, and washed once in 10 mM Tris (pH 8). Libraries were then prepared using Nextera primers, as in the ATAC-seq protocol. Libraries for ChIP-seq were generated using the NEBNext Ultra II DNA library prep kit as per manufacturer's instructions; 10 cycles were used during indexing. Libraries were sequenced paired end on an Illumina NextSeq platform using 500/550 High Output Kit Kit v2.5 (75 Cycles) allocating ~40 million reads per sample.

Data were analysed using an in-house pipeline[64]. PCR duplicates were removed, and biological replicates were normalised to Reads Per Kilobase per Million (RPKM) mapped reads using deepTools package[65], averaged across three biological replicates using bedtools[66], and visualised using University of California Santa Cruz (UCSC) genome browser[67].

### Rad21, Med1 and PolII ChIP coverage quantification

Coverage tracks from Rad21, Med1 and PolII ChIP-seq were generated using deepTools bamCoverage, using single base pair window (--bin-Size 1) and normalized using RPKM (--normalizeUsing RPKM). These bigwig coverage tracks were used as inputs for LanceOtron[68] callpeaks using default settings. The resulting peak regions were filtered for peaks with scores greater than 0.8 and concatenated across samples using bedtools intersect and bedtools merge functions to create a shared peak list with no repeated regions. In addition, fixed sized

regions covering the 5′ and 3′ region of the Hba locus and Hbb locus were also used as inputs for read coverage calculation:

Coordinates used for read coverage calculation, based on an mm39 reference

chr11:32148464-32175527    5′ region covering Snrnp25 and Rhbdf1
chr11:32224569-32251632    3′ region covering Hba-x to Hba-a2
chr7:103479350-103506413 3′ region of Hbb covering Hbb-bh2 to Hbb-y

To determine the read coverage over the regions of interest listed above, Rad21, Med1 and PolII ChIP-seq bam files from each sample were used as inputs for bedtools multicov. Read coverage of peaks in the Hba locus were normalized to peaks in the Hbb locus to correct for sample variation and normalized to E14 WT signals.

## NG Capture-C

Next-generation Capture-C was performed as previously described[28]. A total of $1–2 \times 10^7$ Ter119+ erythroid cells were used per biological replicate. We prepared 3 C libraries using the DpnII-restriction enzyme for digestion. We added Illumina TruSeq adaptors using the NEBNext Ultra II DNA Library Prep Kit for Illumina (New England Biolabs: E7645) according to the manufacturer's instructions, and performed capture enrichment using NimbleGen SeqCap EZ Hybridization and Wash Kit (Roche: 05634261001), NimbleGen SeqCap EZ Accessory Kit v2 (Roche: 07145594001), and previously published custom biotinylated DNA oligonucleotides (R1 and HS-38 viewpoints[21]; α-globin promoters' viewpoints[28]. NG Capture-C data were analysed using the Capture-Compendium toolkit[69] which uses Bowtie[70] to map reads to the mm9 mouse genome build. Cis reporter counts for each sample were normalised to 100,000 reporters for calculation of the mean and standard deviation (three biological replicates). CaptureCompare was used to generate comparisons of the Capture-C interaction profiles (https://github.com/djdownes/CaptureCompare). Briefly, unique interactions were normalised to cis interactions, averaged ($n = 3$), and the difference between the means calculated. For visualisation, means and standard deviations were binned into 150 bp bins and smoothed with a sliding window of 6 kb.

## RNA expression analysis (Real-time PCR and RNA-seq)

Total RNA was isolated from $10^5$-$2 \times 10^6$ Ter119+ erythroid cells lysed in TRI reagent (Sigma-Aldrich: T9424) using a Direct-zol RNA MiniPrep kit (Zymo Research: R2050). DNase I treatment was performed on the column as recommended in the manufacturer's instructions but with an increased incubation of 30 min at room temperature. To assess relative changes in gene expression by qPCR, cDNA was synthesised from 500 ng–1 μg of total RNA using SuperScript III First-Strand Synthesis SuperMix for qRT-PCR (Invitrogen, ThermoFisher: 11752-050) according to the manufacturer's instructions. The ΔΔCt method was used for relative quantification of RNA abundance using Fast SYBR Green Master Mix (Thermo Fisher). Data were normalised to the 18S ribosomal gene or relevant β-globin genes using the ΔCT method, taking the mean of multiple control CT values when applicable. Primers used for the genes assessed are in Supplementary Table 6. Data were subjected to either an unpaired t-test or a one-way or two-way ANOVA and a Tukey post-hoc or a Sidak multiple comparisons test as specified in figure legends. Source data including statistical analysis is available in the Source Data file.

For RNA-seq libraries, 1–2 μg of total RNA was depleted of rRNA and globin mRNA using the Globin-Zero Gold rRNA Removal Kit (Illumina: GZG1224) according to the manufacturer's instructions. To enrich for mRNA, poly(A) + RNA was isolated, strand-specific cDNA was synthesised, and the resulting libraries prepared for Illumina sequencing using the NEBNext Poly(A) mRNA Magnetic Isolation Module (New England Biolabs: E7490) and the NEBNext Ultra II Directional RNA Library Prep Kit for Illumina (New England Biolabs: E7760) following the manufacturer's instructions. Poly(A)+ and Poly(A)- RNA-seq

libraries were sequenced on the Illumina Nextseq platform using a 75-cycle paired-end kit (NextSeq 500/550 High Output Kit v2.5: 20024906). Reads were aligned to the mm9 mouse genome build using STAR[71]. DeepTools bamCoverage was used to calculate normalized (RPKM) and strand-specific read coverage, which was visualized in the UCSC genome browser. Mapped RNA-seq reads were assigned to genes using Subread featureCounts using RefSeq gene annotation. Normalized differential gene expression, between biological triplicate data from littermate wild-type and $SE^{INV}$ mutant mice extracted in parallel, was calculated with the DESeq2 R package.

## Reporting summary

Further information on research design is available in the Nature Portfolio Reporting Summary linked to this article.

## Data availability

All data generated for this study are included in this published article and its supplementary information. Source data are provided as a Source Data file with this paper. ChIP-seq, ATAC-seq, RNA-seq and NG Capture-C data (sequence reads and processed files) is available in the Gene Expression Omnibus (GEO) under accession numbers GSE184435 and GSE211238. In addition to raw and processed files accessible through GEO, NG Capture C data and visualization for the 42 promoter captures are available on this link: https://capturesee.molbiol.ox.ac.uk/projects/capture_compare/3950. Source data are provided with this paper.

## Code availability

Codes used in the analysis of this manuscript have not been specifically written for this paper and are referenced in the relevant assays.

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

## Acknowledgements

This work was supported by UKRI - Medical Research Council (MRC Core Funding. MR/N00969X/1, Programme grant MR/T014067/1), H.S.F. Wellcome Trust Studentship (109097/Z/15/Z), J.B. Wellcome Trust Studentship (219979/Z/19/Z), L.C. Wellcome Trust Studentship (222843/Z/21/Z). D.R.H. is also supported by The Chinese Academy of Medical Sciences (CAMS) Innovation Fund for Medical Science (CIFMS) (grant: 2018-I2M-2-002). M.T.K. is also funded by BBSRC-NSF/BIO (BB/Y008898/1). This work was supported by the Chinese Academy of Medical Sciences (CAMS) Innovation Fund for Medical Science (CIFMS), China (grant number: 2018-I2M-2-002).

## Author contributions

Methodology, M.T.K., A.J.H.S., J.R.H.; Investigation, M.T.K, H.S.F., M.G., M.C.S., C.H., M.L.; Formal Analysis, M.T.K, H.S.F., M.G., M.C.S., C.H., M.L., M.O., L.C., J.B., D.J.D., B.X. N.S.; Software, J.T.; Resources, J.A.S., J.S.S.; Visualisation, M.T.K., H.S.F., D.J.D, M.O., L.C., J.B., Y.S.; Writing-Original Draft, M.T.K., D.R.H.; Writing- Review & Editing, M.T.K., D.R.H., C.B., M.O., D.J.D, H.S.F., A.J.H.S., J.B.; Supervision, D.R.H., J.R.H.; Conceptualization, M.T.K., D.R.H.; Funding Acquisition, D.R.H., M.T.K.

## Competing interests

J.R.H. is a founder and shareholder of Nucleome Therapeutics; J.R.H., and D.J.D. are paid consultants for Nucleome Therapeutics. J.R.H. holds patents for NG Capture-C. Patent applicant: Oxford University Innovation Limited, Name of inventor(s): James R Hughes and James Davies nos. WO2017068379A1, EP3365464B1 and US10934578B2. Specific aspect of manuscript covered in patent application: NG-Capture C experiments. These authors declare no other financial or non-financial interests. The remaining authors declare no competing interests.
