## [Peer Review file · Nature Communications]

The α -globin super-enhancer acts in an orientation-dependent manner

Corresponding Author: Dr Mira Kassouf

Version 0:

Reviewer comments:

Reviewer #4

(Remarks to the Author)

The manuscript "The globin super-enhancer acts in an orientation-dependent manner" by Kassouf et al. reports a detailed analysis of the orientation-dependent effects of a super-enhancer based on a series of structural rearrangements including a 50 kb inversion at the α -globin locus.

First, the authors inverted a single enhancer (R2) within the super-enhancer (SE) and showed that this had no effect on α -globin gene expression or chromatin accessibility, reinforcing the paradigm that enhancers function in an orientation-independent manner. In contrast, when the entire SE was inverted (SEINV), α -globin expression was significantly reduced to ~20% of normal levels, in vitro mESC models and a mouse model. The authors convincingly describe how this inversion re-directs chromatin interactions from the α -globin genes to upstream genes, Rhbdf1 and Snmp25, triggering their upregulation. Moreover, for Rhbdf1, the authors show how the polycomb-repressed marks are replaced with transcription-associated H3K4me3 marks. In addition, they systematically test if two CTCF binding sites and the Mpg gene, included in the inversion but not part of the SE-defined cluster, contribute to the strong effects seen in the SEINV mutants. Further, Kassouf et al. demonstrated functional polarity of the SE by showing asymmetry in cohesin, mediator, and PolII complex distribution along the SE in wild-type. Importantly, this asymmetry is largely maintained for PolII and cohesin ChIP-seq, even when CTCF and Mpg are depleted. These experiments, along with others, show that the SE at the α -globin locus acts in an orientation-dependent manner. Overall, the paper is well-written, with high-quality data that supports the findings through multiple experimental approaches and we are in favor of its publication in Nature Communication.

Major Point

A key limitation in the experimental design, likely addressing some of Reviewer 1's concerns, is that the authors inverted a 50 kb genomic region, which includes not only the 25-30 kb SE but also an additional 15-20 kb region harboring the Sub-TAD boundary and the Mpg gene. The authors explain this chosen inversion with the highly conserved synteny of this region, and potential genotoxicity upon disruptions of Mpg or Nprl3. Since the authors inverted a 50 kb region, including the sub-TAD boundary at the α -globin locus, they later removed two CTCF sites, which partially rescued the altered chromatin interactions, Rad21 loading, and gene misexpression observed in the SEINV mutant. The authors conclude that CTCF removal had little impact.

Results: "We found that the major changes (chromatin state, expression, and interactions) seen in the inversion were not confounded by an insulator effect (comparing data from Δ CTCF to SEINV- Δ CTCF, Figure 5B, C, S8, S9A) nor fully reversed by removing HS38/39 from the inverted allele (comparing SEINV to SEINV- Δ CTCF, Figure S7C). In the absence of HS38/39, the direction in which the SE exerts its effect appears to be encoded within the SE itself."

Discussion: "The original inversion placed CTCF sites (HS38/39) between the enhancers and promoters, but subsequent removal of this site had little effect on α -globin expression."

Although this is in principle correct, the statements may give a misleading impression. The authors should clarify that the observed effects are not solely due to the orientation-dependence of the SE but also involve the CTCF sites. This should be stated in both the results and discussion.

The contribution of the CTCF sites is evident in the following figures:

- Figure S9A:

- o Comparing data from (Δ CTCF to SEINV - Δ CTCF): The displayed difference is likely significant, but no statistical testing for

this comparison is indicated. Statistical testing for Δ CTCF to SEINV- Δ CTCF in S9A should be performed.

o The inversion SEINV leads to a downregulation of α -globin genes to 20% compared to WT. The deletion of CTCF sites HS38/39 significantly rescues this effect (α -globin being expressed at 40% of wild-type levels; comparing SEINV versus SEINV - Δ CTCF).

• Figure 7:

o The authors observe a change in cohesin enrichment at the 5' Hba region (Rhbd1 and Snrnp25 genes) and less cohesin enrichment at the 3' Hba region (α -globin genes) in the SEINV mutant. However, the reduced cohesin enrichment at the 3' Hba region is partially rescued by 50% when the CTCF sites and Mpg are deleted (comparing SEINV versus SEINV - Δ CTCF- Δ Mpg).

• Figure S7:

o Comparing the Capture C profiles shows a loss of interaction between the SE and α -globin genes in the inversion mutant (S7A: comparing WT versus SEINV). However, this effect is partially rescued by removing the CTCF sites (S7C: SEINV versus SEINV - Δ CTCF).

Minor points

1. In Figure 1, the wild-type schematic shows R3 in red rather than blue.
2. The naming of the mutants, INVEN = SEINV and INVEN-KO = SEINV- Δ CTCF, has been inconsistently used in the figures, legends, and text. Please ensure consistent naming throughout to avoid confusion.
3. Could you please check the NG Capture C tracks in Figures 5B and S7C for the SEINV- Δ CTCF? They appear not to fully match. Are these differently merged replicates?
4. In Figure S8, the labeling on the right side appears to be misplaced and does not correspond to the figure.
5. Please spell out "super enhancer" the first time it is mentioned before using the abbreviation "SE."
6. Figure legend 4 is missing the "C."
7. When referring to the statement "did not rescue the phenotype," please specify which phenotype is being referenced for better clarity.

Reviewer #5

(Remarks to the Author)

Version 1:

Reviewer comments:

Reviewer #4

(Remarks to the Author)

We have reviewed the changes made by the authors and are satisfied with their answers and the revisions.

Reviewer #5

(Remarks to the Author)

REVIEWER COMMENTS

Reviewer #4 (Remarks to the Author):

The manuscript "The globin super-enhancer acts in an orientation-dependent manner" by Kassouf et al. reports a detailed analysis of the orientation-dependent effects of a super-enhancer based on a series of structural rearrangements including a 50 kb inversion at the α -globin locus.

First, the authors inverted a single enhancer (R2) within the super-enhancer (SE) and showed that this had no effect on α -globin gene expression or chromatin accessibility, reinforcing the paradigm that enhancers function in an orientation-independent manner. In contrast, when the entire SE was inverted (SEINV), α -globin expression was significantly reduced to ~20% of normal levels, in vitro mESC models and a mouse model. The authors convincingly describe how this inversion re-directs chromatin interactions from the α -globin genes to upstream genes, *Rhbdf1* and *Snrnp25*, triggering their upregulation. Moreover, for *Rhbdf1*, the authors show how the polycomb-repressed marks are replaced with transcription-associated H3K4me3 marks. In addition, they systematically test if two CTCF binding sites and the *Mpg* gene, included in the inversion but not part of the SE-defined cluster, contribute to the strong effects seen in the SEINV mutants. Further, Kassouf et al. demonstrated functional polarity of the SE by showing asymmetry in cohesin, mediator, and PolII complex distribution along the SE in wild-type. Importantly, this asymmetry is largely maintained for PolII and cohesin ChIP-seq, even when CTCF and *Mpg* are depleted. These experiments, along with others, show that the SE at the α -globin locus acts in an orientation-dependent manner. Overall, the paper is well-written, with high-quality data that supports the findings through multiple experimental approaches and we are in favor of its publication in Nature Communication.

Major Point

A key limitation in the experimental design, likely addressing some of Reviewer 1's concerns, is that the authors inverted a 50 kb genomic region, which includes not only the 25-30 kb SE but also an additional 15-20 kb region harboring the Sub-TAD boundary and the *Mpg* gene. The authors explain this chosen inversion with the highly conserved synteny of this region, and potential genotoxicity upon disruptions of *Mpg* or *Nprl3*. Since the authors inverted a 50 kb region, including the sub-TAD boundary at the α -globin locus, they later removed two CTCF sites, which partially rescued the altered chromatin interactions, Rad21 loading, and gene misexpression observed in the SEINV mutant. The authors conclude that CTCF removal had little impact.

Results: "We found that the major changes (chromatin state, expression, and interactions) seen in the inversion were not confounded by an insulator effect (comparing data from Δ CTCF to SEINV- Δ CTCF, Figure 5B, C, S8, S9A) nor fully reversed by removing HS38/39 from the inverted allele (comparing SEINV to SEINV- Δ CTCF, Figure S7C). In the absence of HS38/39, the direction in which the SE exerts its effect appears to be encoded within the SE itself."

Discussion: "The original inversion placed CTCF sites (HS38/39) between the enhancers and promoters, but subsequent removal of this site had little effect on α -globin expression." Although this is in principle correct, the statements may give a misleading impression. The authors should clarify that the observed effects are not solely due to the orientation-

dependence of the SE but also involve the CTCF sites. This should be stated in both the results and discussion.

The contribution of the CTCF sites is evident in the following figures:

- Figure S9A:

- o Comparing data from (Δ CTCF to SEINV - Δ CTCF): The displayed difference is likely significant, but no statistical testing for this comparison is indicated. Statistical testing for Δ CTCF to SEINV- Δ CTCF in S9A should be performed.

- o The inversion SEINV leads to a downregulation of α -globin genes to 20% compared to WT. The deletion of CTCF sites HS38/39 significantly rescues this effect (α -globin being expressed at 40% of wild-type levels; comparing SEINV versus SEINV - Δ CTCF).

- Figure 7:

- o The authors observe a change in cohesin enrichment at the 5' Hba region (*Rhbdf1* and *Snrnp25* genes) and less cohesin enrichment at the 3' Hba region (α -globin genes) in the SEINV mutant. However, the reduced cohesin enrichment at the 3' Hba region is partially rescued by 50% when the CTCF sites and Mpg are deleted (comparing SEINV versus SEINV - Δ CTCF- Δ Mpg).

- Figure S7:

- o Comparing the Capture C profiles shows a loss of interaction between the SE and α -globin genes in the inversion mutant (S7A: comparing WT versus SEINV). However, this effect is partially rescued by removing the CTCF sites (S7C: SEINV versus SEINV - Δ CTCF).

Our response in blue:

The reviewer makes a good point. The reporting of the results and wording of the conclusions should better reflect the observed data.

We have therefore changed the wording following the reviewers' advice, as you will see in 'Tracked changes' both in the results section and discussion. In the results section, by adding the expression 'not completely' and adverb 'partly', we emphasise the novel finding (SE polarity) but do not claim that it's solely responsible for the phenotype. In fact, in the conclusion of that section we do say "In summary, re-positioning of the HS3839 element is not the only cause for reduced interaction between the SE and the α -globin genes nor it is solely responsible for the perturbed expression of the surrounding genes, including the upregulation of *Rhbdf1* and *Snrnp25* genes." acknowledging the CTCF undeniable role in the inversion phenotype but without taking away from the new finding.

In the discussion we modified the sentence to the following version:

"The original inversion placed CTCF sites (HS3839) between the enhancers and promoters, but subsequent removal of these sequences clarified the distinct effects both the CTCF sites and the enhancer inversion had on the downregulation of α -globin expression."

Minor points

1. In Figure 1, the wild-type schematic shows R3 in red rather than blue.

Corrected

2. The naming of the mutants, INVEN = SEINV and INVEN-KO = SEINV- Δ CTCF, has been inconsistently used in the figures, legends, and text. Please ensure consistent naming throughout to avoid confusion.

Reviewed and corrected in figures, legends, and text.

3. Could you please check the NG Capture C tracks in Figures 5B and S7C for the SEINV- Δ CTCF? They appear not to fully match. Are these differently merged replicates? These are the same merged triplicates that are put through the differential analysis pipeline. Unfortunately, the conversion of the figures I had produced a while back into vectors in Illustrator reduces the resolution on this set of data. I no longer have the original tracks and would need to reanalyse the data using new pipelines to recreate these comparison files. Since this is supplementary data and it does not reveal additional insights, I could delete it if unsatisfactory or leave it as is, whatever you advise.

4. In Figure S8, the labeling on the right side appears to be misplaced and does not correspond to the figure.

Corrected

5. Please spell out "super enhancer" the first time it is mentioned before using the abbreviation "SE."

Corrected

6. Figure legend 4 is missing the "C."

Corrected

7. When referring to the statement "did not rescue the phenotype," please specify which phenotype is being referenced for better clarity.

This sentence is in the discussion section. It was replaced by the following sentence: "Importantly, the model in which both the CTCF site and the *Mpg* gene were removed did not restore the α -globin expression and the accompanying chromatin state changes nor did it attenuate the upregulation of the genes located at the 5' of the α -locus."

Reviewer #5 (Remarks to the Author):

Thank you for this thorough and very helpful review.